# Wnt Pathway Extracellular Components and Their Essential Roles in Bone Homeostasis

**DOI:** 10.3390/genes13010138

**Published:** 2022-01-13

**Authors:** Núria Martínez-Gil, Nerea Ugartondo, Daniel Grinberg, Susanna Balcells

**Affiliations:** Department of Genetics, Microbiology and Statistics, Faculty of Biology, Universitat de Barcelona, CIBERER, IBUB, IRSJD, 08028 Barcelona, Spain; nereaugartondo@gmail.com (N.U.); dgrinberg@ub.edu (D.G.); sbalcells@ub.edu (S.B.)

**Keywords:** Wnt pathway, inhibitors, ligands, co-receptors, bone

## Abstract

The Wnt pathway is involved in several processes essential for bone development and homeostasis. For proper functioning, the Wnt pathway is tightly regulated by numerous extracellular elements that act by both activating and inhibiting the pathway at different moments. This review aims to describe, summarize and update the findings regarding the extracellular modulators of the Wnt pathway, including co-receptors, ligands and inhibitors, in relation to bone homeostasis, with an emphasis on the animal models generated, the diseases associated with each gene and the bone processes in which each member is involved. The precise knowledge of all these elements will help us to identify possible targets that can be used as a therapeutic target for the treatment of bone diseases such as osteoporosis.

## 1. Introduction

The Wnt pathway (wingless-type mouse mammary tumor virus integration site) is a ubiquitous pathway involved in a wide variety of cellular processes, and highly conserved across species. It was first described for its role in carcinogenesis, and later demonstrated to be essential for embryonic patterning and for maintaining adult tissues [1,2,3]. In the skeletal tissue, the Wnt pathway is involved in the differentiation of osteoblasts, osteoclasts and chondrocytes. Therefore, it has a very important role in both skeletal development and bone remodeling, and an essential function in mechanical communication.

Historically, the Wnt pathway has been categorized into β-catenin-dependent (canonical pathway) and β-catenin independent (non-canonical) pathways (Figure 1). Although some studies postulate that different Wnts activate either the canonical or the non-canonical pathways, it has also been reported that some Wnt ligands (e.g., WNT5A) can act on both pathways [4,5]. In all cases, the Wnt ligands bind to one of the 10 frizzled family receptors (FZD) and optionally to a co-receptor that can be from the low-density lipoprotein receptor-related protein (LRP) family (LRP5/6) or a transmembrane tyrosine kinase (ROR1/2 or RYK) [6,7,8]. The FZDs are seven-pass transmembrane G-protein receptors that can transmit signals through dishevelled proteins or through heterotrimeric G proteins [9,10]. Although the canonical pathway has received much attention, there is increasing evidence on the role of the non-canonical pathways in bone and the crosstalk between the pathways [11,12,13]. The canonical Wnt pathway begins with the formation of a heterotrimeric complex consisting of an FZD, a Wnt ligand and an LRP co-receptor (Figure 1). This binding results in the stabilization and translocation of β-catenin into the nucleus, where it activates the transcription of important target genes. When the pathway is not active, the ‘destructive complex’ phosphorylates and sequesters β-catenin in the cytoplasm for its subsequent degradation in the proteasome. The canonical Wnt pathway is regulated by a series of extracellular inhibitors common to all the Wnt pathways (e.g., secreted frizzled-related proteins (SFRP) and Wnt Inhibitory factor 1 (WIF-1)) but also by specific canonical Wnt pathway inhibitors that block the LRP5/6 binding (e.g., sclerostin and DKK1) [12,13]. In addition to the canonical pathway, Wnt ligands can activate multiple signaling cascades independent of β-catenin. Within this category, the most widely studied are the WNT-planar cell polarity (WNT-PCP) and the WNT-calcium (WNT-Ca^2+^) pathways (Figure 1) [14,15,16,17], which can branch out into the JNK, PI3K-RAC1 and mTORC1/2 pathways, among others [18,19,20,21,22]. The WNT-PCP pathway begins with the binding of a Wnt ligand, such as WNT5A, to FZD and to ROR1/2 or RYK co-receptors. This binding activates small G proteins, such as Rac and Rho, which are implicated in the establishment of cell polarity and cell migration [23]. In addition, the activation of these pathways may inhibit the canonical Wnt pathway [24,25]. The WNT-Ca^2+^ pathway regulates calcium release from the endoplasmic reticulum in order to control intracellular calcium levels. It begins with the binding of a Wnt ligand, such as WNT5A, to FZD, which in turn activates the phospholipase C (PLC). Cell adhesion, migration and embryonic development are regulated through this pathway [26].

Despite the fact that the Wnt pathways are some of the most studied signaling routes, their complexity means that there are still many aspects to discover. To precisely regulate the transcription of Wnt pathway target genes, complex and tight regulation is carried out by extracellular components including ligands (19 Wnt ligands), inhibitors (12 different: SFRPs, Dickkopf (DKKs), sclerostin, among others), co-receptors (LRP, ROR, among others) and receptors (10 FZD). A current challenge is determining the tissue-specific and transient modulators of this pathway—for instance, in bone tissue—which could be useful as new therapeutic targets. Importantly, many rare and complex diseases are associated with genes from members of the Wnt pathway (Table 1). Since treatments for bone density diseases target genes that cause rare monogenic bone diseases, a comprehensive understanding of the causal variants and genes is fundamental.

This review aims to describe, summarize and update the findings regarding all these extracellular modulators in relation to bone homeostasis, with an emphasis on the animal models generated, the diseases associated with each gene and the bone processes in which each member is involved. The review is structured into three main sections—co-receptors, ligands and inhibitors—within which we will review the main members of the Wnt pathway that have an important role in bone homeostasis.

## 2. Co-Receptors

We define a co-receptor as a cell surface receptor that binds to both a signaling molecule and to a primary receptor, thereby facilitating ligand recognition and initiation of downstream biological processes. Different co-receptors play essential roles in Wnt signaling, both in the canonical and non-canonical pathways. This section includes a subsection entitled LRPs, with information related to LRP4/5/6; a second subsection entitled Kremen, which includes information on Kremen1/2; a third subsection, ROR, which includes information related to ROR1/2, and a final subsection, LGR, which includes information related to LGR4/5/6. We review the role of these co-receptors in bone remodeling, with a special emphasis on mutations causing disease (Figure 2) and on the animal models available (Table 2).

### 2.1. LRPs

The LRP proteins belong to the low-density lipoprotein receptor (LDLR) family of highly conserved cell surface receptors [117], whose functions include endocytosis (e.g., lipoproteins or proteases) and direct signal transduction (e.g., bone morphogenic protein (BMP) or Wnt pathways) [118,119]. These transmembrane proteins have a large extracellular domain (ECD) and a small intracellular domain (ICD) separated by a single-pass transmembrane domain. The ECD allows interaction with extracellular proteins (e.g., Wnt ligands or extracellular inhibitors such as DKK1 or sclerostin), and the ICD is in charge of the downstream transmission of signaling events [120].

The most studied LRP family members in relation to bone are LRP5 and LRP6, which share 71% homology at the protein level [121]. Their ECDs consist of four β-propeller domains (class B repeats or YWTD domains) alternating with four epithelial growth factor (EGF)-like domains followed by three LDLa domains (class A repeats). Their ICDs include various highly conserved PPPS/TP motifs containing serines and threonines that are phosphorylated upon receptor activation (Figure 2). These phosphorylated sites bind Axin, thereby preventing the phosphorylation of β-catenin [7,122,123]. LRP4, also known as MEGF7, has a structure similar to that of LRP5/6 but with some differences. The ECD features eight LDLa domains, four β-propeller domains alternating with six EGF-like repeats and a domain for O-linked oligosaccharide modification. The ICD features two domains: the endocytosis signal NPxY domain responsible for protein internalization, and the PZD-interacting motif at the C-terminal end (Figure 2) [124,125,126].

The importance of these transmembrane co-receptors in bone was revealed by the description of mutations in *LRP5* that caused two diametrically opposed bone phenotypes: osteoporosis pseudoglioma syndrome (OPPG), and the high bone mass (HBM) phenotype [38,50,51] (Table 1 and Figure 2). The difference between the two types of mutations is that OPPG results from loss-of-function (LoF) mutations causing *LRP5* to be unable to activate the canonical Wnt pathway. In contrast, the mutations responsible for the HBM phenotype are gain-of-function (GoF) and cause a loss of affinity for the extracellular inhibitors DKK1 and sclerostin. In this case, LRP5 can no longer be internalized and remains available in the membrane for Wnt pathway activation [39,127,128,129,130,131,132]. In addition, the *LRP5* locus has been associated with bone mineral density (BMD) and risk of fracture in a plethora of genome-wide association studies (GWAS) [133,134,135,136,137,138,139,140,141,142,143].

To understand the effect of LRP5 on bone and the signaling pathways by which it acts, genetically modified mouse models have been generated (Table 2). Lrp5 total knock-out (Lrp5-KO) mouse models and conditional KOs (cKOs) in bone-related cells do not display bone alterations at birth but acquire a decreased BMD during postnatal development due to reduced bone formation [80,81,83,84,85,86,87,88,89] (Table 2). On the contrary, global or conditional LRP5 transgenic mice carrying the human LRP5 HBM p.Gly171Val or p.Ala214Val mutations reproduce the HBM phenotype with high rates of bone formation and better bone quality [88,90] (Table 2). Furthermore, conditional knock-in (cKI) female mice with the HBM mutations present only in osteoclasts show a reduction in bone resorption, demonstrating that the in vivo effect of the HBM mutations is not only due to osteoblast stimulation but also to osteoclast inhibition [91] (Table 2). Relative to skeletal mechano-responsiveness, Lrp5-KO and Lrp5-cKO mice have been reported to show severe impairment in later stages of the mechano-transduction signaling cascade, whereas HBM mice show a higher osteogenic response to mechanical loading in vivo than WT mice [84,144,145]. In 2008, Yadav et al. proposed that the effect of LRP5 on bone formation and BMD was due to its inhibitory effect on tryptophan hydroxylase 1 and, therefore, on serotonin synthesis in the duodenum, which would otherwise block bone formation by signaling in osteoblasts [82,146,147]. However, these hypotheses have not been reproduced, and there is much evidence that LRP5 exerts its effect directly on osteoblastic cells through the Wnt canonical pathway [86,87,88,148,149].

Similar to LRP5, there is much evidence of a role of LRP6 in bone. Missense mutations in *LRP6* have been described in humans as the cause of tooth agenesis and oligodontia and, recently, HBM [53,62,63] (Table 1 and Figure 2). Interestingly, the mutations causing the HBM phenotype are located exclusively in the first β-propeller domain, and the phenotype of these patients is identical to that of the patients with LRP5-HBM. Moreover, *LRP6* has been associated with heel BMD in three different studies [140,141,142].

The Lrp6-KO mouse model is embryonic lethal with loss of distal limb structures and truncation of the axial skeleton, indicating that Lrp6 plays a fundamental role during embryogenesis, during which it is widely expressed [83,86,92] (Table 2). Confirmation that Lrp6 is also involved in postnatal bone metabolism was obtained using cKO mouse models and models with hypomorphic mutations [87,89,93,94] (Table 2). Specifically, the Ringelshwanz (rs) mouse with the p.R886W hypomorphic mutation in the third β-propeller domain showed developmental malformations and delayed ossification, in addition to an increase in bone resorption and a decrease in BMD and in the activity of the canonical Wnt pathway in osteoblasts [93,94] (Table 2). Through different in vitro models, it was determined that this effect was a consequence of the impaired interaction of mutant Lrp6 with the chaperone Mesd and its defective trafficking to the plasma membrane [93,94]. Lrp6-cKO mice in the early osteoblast lineage showed reduced BMD due to decreased bone formation [87,89]. In contrast, there was no obvious bone phenotype for Lrp6-cKO in the embryonic mesenchyme [86] (Table 2).

The double KO and cKO for both co-receptors (Lrp5 + Lrp6-dKO and Lrp5 + Lrp6-cdKO) show a more severe phenotype than the single KOs and cKOs regarding BMD and skeletal development during embryogenesis (Table 2). Many authors agree that there is some redundancy in the functions of these co-receptors, but that they may regulate the Wnt pathway in different time windows (Lrp6 in early stages and Lrp5 in the late stages of osteoblast differentiation) and in different skeletal compartments (Lrp6 in trabecular bone and Lrp5 in cortical bone) to promote the proper acquisition of BMD. Therefore, the expression of the two co-receptors in osteoblasts would be necessary for normal skeletal homeostasis [83,86,87,89,150] (Table 2). More research is needed to better define the tissue distribution and the time at which these two co-receptors act. In this context, a study of Lrp5 + Lrp6-cKO at early and late stages of osteoclast differentiation showed that there was a low BMD with reduced osteoblast and osteoclast numbers in the early osteoclast model, while the late osteoclast model did not have a bone phenotype [95] (Table 2).

The characterization of all these animal models indicates that Lrp5 and Lrp6 exert their function in bone homeostasis mostly through the regulation of bone formation, controlling the differentiation and proliferation of osteoblasts, bone matrix deposition in differentiated osteoblasts and apoptosis of osteocytes. Furthermore, these effects are mainly due to Wnt signaling through the canonical pathway. It has been determined that LRP6 acts not only through the formation of the heterotrimeric complex LRP6-FZD-Wnt, but also through the formation of a complex involving LRP6, parathyroid hormone (PTH) and PTH receptor 1, which promotes the stabilization of β-catenin and thus activates the canonical Wnt pathway [151].

LRP4 has been extensively studied in relation to its role in the neuromuscular junction, where it serves as a co-receptor for motor neuron-derived agrin/Wnt and facilitates acetylcholine receptor clustering on the myocyte [152]. In addition to this function, LRP4 has a very important role in bone homeostasis, where it enhances the inhibitory function of sclerostin in the canonical Wnt pathway [47,96,99,100,153]. LRP4 can also enhance other inhibitors of the Wnt pathway, such as DKK1 and SOSTdc1 [99,154]. 

Missense and splicing mutations in *LRP4* have been implicated in several musculoskeletal human diseases depending on the position of the mutation (Table 1 and Figure 2). Mutations in the LDLa, EGF-like domain or in the first and second β-propeller domains cause Cenani–Lenz Syndrome (CLS) [67]; mutations in the central cavity of the third β-propeller domain cause sclerosteosis-2 [47,48]; and those in the fourth β-propeller domain or in the EGF-like domain cause isolated bilateral syndactyly [68,69]. Thanks to in vitro functional studies, it has been proven that all these mutations cause a significant decrease in the inhibition of the canonical Wnt pathway by LRP4, this effect being more severe in mutations associated with sclerosteosis-2. CLS-causing mutations decrease the levels of LRP4 available in the membrane, while this is not observed for the sclerosteosis-causing mutations. Instead, impaired binding between sclerostin and LRP4 has been verified in the mutations causing sclerosteosis, which would explain the severity of the effect [48,67,68]. Interestingly, mutations in the edge of the third β-propeller domain have been found to cause myasthenia gravis, the most common neuromuscular junction pathology. These mutations cause impairment of the MuSK signaling pathway, whereas sclerosteosis-2 mutations have no effect on this pathway, demonstrating the highly position-specific effect of the different LRP4 mutations [155]. Furthermore, as with *LRP5/6*, *LRP4* has been associated with BMD in different GWAS studies [136,137,140,141,142].

Lrp4-KO or null mutant mouse models are not viable due to the absence of neuromuscular junctions and therefore the inability to breathe [96,97,98] (Table 2). In contrast, Lrp4-cKO mouse models in the osteo lineage are viable. When the deletion occurs specifically in osteoblasts or in osteocytes, a high BMD is observed, with high levels of bone formation and reduced bone resorption but without polysyndactyly [96,100] (Table 2). The phenotype is more severe in early osteoblasts, where Lrp4 is more highly expressed, than in osteocytes [100]. In contrast, no bone phenotype is observed in osteoclast-specific KO mice [96] (Table 2). The presence of Lrp4 has been shown to be essential in these animal models for the functioning of sclerostin [96]. In 2017, Xiong et al. [156] proposed that reduced bone resorption in Lrp4-cKO in the osteoblast lineage stabilizes the prorenin receptor and, as a result, increases the production and secretion of the ATP derivative adenosine. Elevated adenosine-A2aR signaling in osteoclast precursors reduces receptor activator of nuclear factor kB (RANK)-mediated osteoclastogenesis. Furthermore, circulating sclerostin levels are higher in Lrp4-cKO in the osteoblast lineage with unchanged expression of *SOST* per cell, but it is possible that increasing the number of osteocytes with increasing bone formation will also increase the total level of *SOST*. In addition to these models, LRP4-KI mouse models with the p.R1170W or p.R1170Q mutations, which cause sclerosteosis in humans, recapitulate the human HBM phenotype associated with this mutation without the presence of syndactyly (Table 2). In addition to high BMD, these mice exhibit high levels of bone formation with unchanged bone resorption and are protected against the effects of anti-sclerostin antibodies and of *SOST* transgenic overexpression [101,102]. Polysyndactyly and other limb deformity problems have been reported in different murine and bovine hypomorphic models of Lrp4 [98,99,153,154,157,158,159]. Specifically, the Lrp4 ECD mouse model that does not have the transmembrane or intracellular domains has polysyndactyly and decreased BMD [99] (Table 2).

All these studies on animal models, together with many in vitro studies, seem to indicate that Lrp4 exerts its function on the differentiation/function of osteoblasts through the inhibition of the Wnt-induced activation of the canonical pathway. 

### 2.2. ROR

ROR1/2 belong to the evolutionarily conserved Ror family of receptor tyrosine kinases, which are type I transmembrane protein tyrosine kinases. These receptors contain different motifs, including an immunoglobulin-like domain, a frizzled-like cysteine rich domain (CRD) and a kringle domain (KD) in the extracellular region, a transmembrane section and an intracellular region with a tyrosine kinase domain, two serine/threonine-rich domains, a proline-rich domain and a short C-terminus tail [160,161,162] (Figure 2). There is some controversy about the activity of these receptors, regarding whether they are true kinases or pseudokinases [163]. ROR1 and ROR2 are structurally very similar, with 58% amino acid identity and high domain similarity. Both bind WNT5A through their CRD domains, and, by acting as receptors or co-receptors along with FZD, they activate the non-canonical Wnt signaling pathway [8,163]. The Ror1 + Ror2-dKO mouse model revealed that these two co-receptors possess certain redundant functions in skeletal and cardiac development [164,165]. Ror1/2 are widely expressed in various tissues and organs, such as the limbs, heart, lungs, gut, muscles and skeletal and nervous systems, during embryonic development, while their expression decreases markedly in adult tissues [163]. While Ror2 is strongly expressed in mature osteoclasts, Ror1 expression has not been detected in these cells [107]. In addition, Ror2 has been reported to regulate the length of various organs, including limbs, through interaction with both Wnt5a and Wnt9a [163,166]. 

In humans, recessive LoF mutations in *ROR2* are associated with autosomal recessive Robinow syndrome, and dominant GoF mutations are associated with brachydactyly type B [57,58,70,107,160] (Table 1 and Figure 2). Moreover, *ROR2* has been associated with hip BMD in a GWAS study [164].

Ror2-KO mouse models show early lethality and abnormalities characteristic of Robinow patients. The Ror2-KO mice exhibit dwarfism, facial abnormalities, short and deformed limbs and tails, dysplasia of lungs and genitals, ventricular septal defects and severe cyanosis [8,103,104,105] (Table 2). In these mice, the bones formed through endochondral ossification are affected, but not those formed by intramembranous ossification. The heterozygous Ror2-KO mouse and the Ror2-cKO in the osteoclast lineage exhibited a HBM phenotype due to impaired bone resorption only in trabecular bone [103,106,107] (Table 2). The impaired bone resorbing activities of osteoclasts can be restored by the constitutively active form of RhoA, showing that this impaired bone resorption is an effect of the non-canonical Ror2-Wnt5a signaling pathway [103,106,107,167]. Thus, Ror-2-mediated signaling regulates the differentiation and bone resorbing activity of osteoclasts, thereby maintaining bone mass. 

### 2.3. KREMEN

The Kremen family of receptors (KRM1/2) are single-pass transmembrane proteins that show high affinity for DKK1 and DKK2 [168,169]. KRM presents an ECD that contains a KD, a carbohydrate binding (WSC) and a CUB domain that are necessary for binding to DKK [168,170] (Figure 2). KRM1/2 can bind to the CRD2 domain of DKK1 and form the DKK1-KRM-LRP5/6 ternary complex, which is internalized, thus removing LRP5/6 from the cell surface [171,172] (Figure 1). Similar to LRP4, which enhances sclerostin inhibitory activity, Kremen acts by enhancing DKK1 function, but unlike LRP4, Kremen is not essential for DKK1 to perform its function [168]. While Krm1 has a more widespread expression pattern, Krm2 is expressed predominantly in bone [109]. In addition to this effect, Krm1/2 can activate the Wnt pathway by binding directly to Lrp6 in the absence of Dkk1 [109,173,174].

In humans, mutations in *KRM1* affect different protein domains and have been associated with ectodermal dysplasia and oligodontia [73,74] (Table 1 and Figure 2).

Krm1-KO mice and young Krm2-KO mice (less than 24 weeks old) are viable without abnormalities and show normal bone formation and bone mass [108], while adult (24 weeks) Krm2-KO mice display higher bone formation and bone mass [108,109] (Table 2). Krm1 + Krm2-dKO are also viable, but show ectopic postaxial forelimb and expanded apical ectodermal ridges and increased bone formation, bone mass and canonical Wnt signaling [108] (Table 2). The dKO mouse phenotype shows similar defects in the limb to those described in the heterozygous Dkk1-KO mutants (see Section 4). In addition, the ectopic growth of digits present in the Krm1 + Krm2-dKO mice is further enhanced by the deletion of a single Dkk1 allele, but without any additional increase in BMD [108]. In contrast, mice with osteoblast-specific Krm2 overexpression display a severe osteoporotic phenotype caused by decreased bone formation and increased bone resorption with impaired fracture healing [109,175] (Table 2). Taking all these models together, it seems that Krm1 and Krm2 show a certain degree of redundancy in skeletal growth and development and that Krm2 is important for bone formation during skeletal maintenance/remodeling.

### 2.4. LGR

The leucine-rich repeat-containing G-protein-coupled receptors (LGRs) 4, 5 and 6 constitute a subfamily of receptors belonging to the G-protein-coupled receptor (GPCR) superfamily [176]. Structurally, these transmembrane GPCRs belong to the class A rhodopsin-like family [177], which have a characteristic leucine-rich domain (LRR) in the extracellular N-terminal region that is responsible for ligand interaction [177,178,179,180]. All three receptors show some homology, but while LGR4/5 have 17 LRR repeats, LGR6 only has 13 [179]. These receptors bind to R-spondins (RSPOs), causing the formation of the LGR-RSPO-ZNRF/RNF43 complex. This leads to the sequestration of ZNF/RNF43, which causes an increase in membrane-available FDZs, since the transmembrane ubiquitin ligases ZNF/RNF43 are responsible for degrading them [31]. In addition to this agonistic effect on the Wnt pathway, LGR is a target gene of the Wnt pathway, which will produce a positive feedback loop [31]. It has also been reported that LGR proteins mark different adult stem cells and play a very important role in defining progenitor and stem cell behavior [31,181,182]. Specifically, it has been reported that LGR5 is a marker of adult stem cells in the stomach [183], hair follicles [184] and small intestine and colon [181], among others, and LGR6 in the taste buds, lungs and skin [115,181,183,185].

LGR4, also known as GPR48, is widely expressed in multiple tissues from early embryogenesis to adulthood [186], being abundantly expressed in skeletal, adipose and muscular tissue [110,113,178,187]. Specifically, expression has been observed in osteoblast precursors in in vitro osteogenesis, and in osteoclasts in response to RANK ligand (RANKL)-induced osteoclast differentiation. In addition to the Wnt pathway enhancement, LGR regulates a multitude of pathways through its classical G-protein signaling potential [110,188]. In one of these pathways, LGR4 acts as a secondary receptor for RANKL, competing with RANK. Furthermore, this LGR4-RANKL binding is inhibited by both osteoprotegerin (OPG) and RSPO1. Through in vitro and in vivo studies, LGR4 has been shown to inhibit RANKL-induced osteoclast differentiation, survival and function [112].

No rare human diseases associated with *LGR4* have been found, but a rare nonsense variant has been linked to osteoporosis in two different association studies [189,190] and common variants are associated with heel BMD [140,142]. Total Lgr4-KO mice are at risk of perinatal death, with approximately 60% of newborn mice dying before day 1. These mice exhibit decreased body weight, size and bone length [110,111,112,113] (Table 2). Lgr4-KO mice have a low BMD due to both impaired bone formation and bone resorption. Regarding bone formation, these mice present a dramatic delay in osteoblast differentiation and mineralization during embryonic bone formation, and, in postnatal bone remodeling, there is a decrease in the kinetic indices of bone formation rate and osteoid formation [110,111]. During bone resorption, they present osteoclast hyperactivation with an increase in the number of osteoclasts, surface area, size and bone erosion [110,112] (Table 2), a phenotype that is also observed in Lgr4-cKO mice in monocytes [112] (Table 2). Interestingly, the injection of a soluble form of the Lgr4 extracellular domain abrogated RANKL-induced bone loss in three mouse models of osteoporosis [112]. Thanks to in vitro and in vivo studies, it has been verified that RSPO signaling has no effect on osteoclast differentiation and that the Lgr4 effect in this cell type is determined by its signaling via RANKL [112,191]. In addition to the evidence from in vivo studies, many in vitro studies have been carried out demonstrating the positive effect of LGR4 on osteogenic differentiation [111,192,193], adipocyte and myocyte differentiation [111]. Although LGR4 is expressed within bone-marrow-derived mesenchymal stem cells (BMSCs) undergoing osteogenic differentiation, its expression is not correlated with any specific osteogenic marker but is maintained throughout the process, suggesting that LGR4 may be necessary to support osteogenic differentiation [194].

LGR5 is also known as GPR49, HG38 and FEX, and is expressed in osteoblast precursors in osteogenesis in vitro. Lgr5-KO mice show neonatal lethality (during the first 24 h of life) associated with ankyloglossia, characterized by the fusion of the tongue to the floor of the mouth, leading to the inability to nurse and subsequent neonatal mortality. Perinatal lethality is due partly to respiratory failure as the result of pressure against the diaphragm from the markedly distended abdomen [114] (Table 2). This phenotype suggests the involvement of LGR5 in craniofacial development. Regarding *LGR6*, as with *LGR4*, a rare variant associated with the risk of postmenopausal osteoporosis has been described [189], along with common variants associated with heel BMD in a GWAS study [140]. It is expressed in mouse primary calvarial cells, bone marrow cells and BMSCs [194]. In vitro studies have highlighted the importance of LGR6 in promoting osteogenic differentiation, correlating its expression with typical markers of osteogenic differentiation, such as osterix [194,195]. These results point to LGR6 as a novel marker of osteoprogenitor cells in bone marrow [194]. A recent study shows that the enhancement of the Wnt pathway by LGR6 occurs downstream of the BMP signaling pathway [196]. Despite this, Lgr6-KO mice develop normally and no skeletal phenotype is evident, although Lgr6 is necessary for the regeneration of the digit tip bone following amputation in mice [115,116] (Table 2).

## 3. WNT Ligands

Wnt ligands are highly conserved, secreted glycoproteins with a molecular weight around 40 KDa. They are formed by 350–400 amino acids, 23–24 of which are conserved cysteine residues. For secretion of an active Wnt ligand, Porcn-mediated lipidation by palmitic acid is necessary and it occurs on a conserved serine residue (Figure 3). Following lipidation, secretion into the extracellular matrix requires Wls binding [32,33,197]. However, some authors have questioned the essential nature of this step [33,198,199]. Although the degree of sequence identity between some Wnt family members is only 18%, it is thought that all Wnt proteins form a similar three-dimensional structure, which is shaped similarly to a fist with a cleft inside and thumb and index finger protrusions [197,198,200]. There are four main regions of concentrated amino acid conservation that map onto the Wnt structure: the tips of the thumb and index finger loops, the core of the N-terminal α-helical domain (NTD) and a large continuous patch in between, facing the exterior. The two main contact points between the Wnt ligand and the FZD receptor are the tips of the thumb and index loops [197,198]. The cysteine-rich domain (CRD) at the second point of contact with the receptor is similar to that of SFRP and ROR1/2, among others [201]. On the NTD, the lipidation of the conserved serine creates hydrophobic forces with the receptor that, combined with shape complementarity, facilitate binding [198]. The other conserved regions of the Wnt proteins, located opposite the FZD binding region, bind to co-receptors such as LRP5/6/RYK [197,198,202]. The amino acid differences between Wnt ligands enable their differential binding to receptors. For example, WNT1 binds to the first β-propeller domain of LPR5/6, whereas WNT3A binds to the third β-propeller domain [203]. It has been suggested that the activation of one or another Wnt pathway relies on the available receptors [204,205], while other studies have suggested that the selection of the pathway depends on the amount of Wnt ligand available [5,206]. A total of 19 Wnt ligands have been identified in humans. Of these, WNT1, WNT3A, WNT4, WNT5A, WNT5B, WNT7A, WNT7B, WNT9A, WNT10A, WNT10B and WNT16 have been extensively studied in relation to their role in bone, both in its function, in those mutations associated with human diseases (Figure 3), and in the animal models generated (Table 3). 

### 3.1. WNT1

WNT1, also known as INT1, plays an important role in osteoblastogenesis and can also inhibit osteoclastogenesis and chondrogenesis. WNT1 is expressed in mesenchymal stem cells (MSCs), osteoblasts, monocytes and osteocytes, in which it can activate the canonical Wnt pathway in an autocrine or paracrine way, and also the mTORC1 pathway in osteoblasts and the PCP-JNK pathway in osteoclasts [211,212,230,241]. Activation of the canonical pathway by WNT1 inhibits MSC differentiation into adipocytes, promoting osteoblastogenesis [207,230]. At a later stage, WNT1 activation of mTORC1 in osteoblasts promotes their differentiation and mineralization [212]. Activation of JNK-dependent non-canonical signaling by WNT1 seems to contribute to the suppression of osteoclast differentiation [207]. In adult bone, WNT1 expression can be induced by mechanical loading or physical activity to activate osteoblastogenesis [242,243,244]. Importantly, the extent of WNT1 induction declines with aging and may underlie the muted bone anabolic response to mechanical loading in aged mice [244].

*WNT1* has been linked to bone physiology in humans, as LoF mutations can cause early-onset osteoporosis or osteogenesis imperfecta (OI) [35,36,40] (Table 1 and Figure 3). The differential diagnosis between early-onset osteoporosis and OI has turned out to be a difficult task, given their relatedness and the genotypic and phenotypic variability within each clinical entity. Lately, they have been considered to be part of a spectrum rather than two delimited disorders. Moreover, *WNT1* is associated with total-BMD and heel-BMD [136,140,141,142,235].

Due to the importance of Wnt1 in the early stages of embryonic development of the nervous system, total Wnt1-KO mouse models resulted in perinatal death [208,241] (Table 3). The heterozygous Wnt1-KO models suffered mild osteopenia but no significantly impaired bone resorption or formation, although osteoblast differentiation and activity were impaired in vitro [207]. The hypomorphic Wnt1 Swaying mouse model has a better neonatal survival rate than the total Wnt1-KO and it recapitulates several phenotypes of OI patients, such as spontaneous fractures and severe osteopenia, caused by a decrease in osteoblast activity (Table 3) [209]. Similarly, the Wnt1-cKO in MSCs [212] and the Wnt1-cKO in late osteoblasts and osteocytes [212] also recapitulate the OI phenotype, with impaired osteoblast activity, decreased mineralization and bone formation rate and spontaneous fractures, the latter displaying a milder phenotype, with conserved osteoblast and osteoclast numbers [207,212] (Table 3). In contrast, when Wnt1 is overexpressed in osteoblasts or osteocytes, a HBM phenotype is obtained [212] (Table 3). The osteoblast number and mineral apposition rate are increased while the osteoclast number per bone surface remains similar to controls. Inhibition of mTORC1 in the Wnt1 GoF model was associated with reduced mineralization and bone formation [212]. Likewise, the OI phenotype of the hypomorphic Wnt1 Swaying mice was also corrected by the mTORC1 activation provoked by Tsc1-cKO in late osteoblasts and osteocytes [212]. In a recent study, a comparison between Wnt1-cKO in osteoblasts and cKO in monocytes showed the development of general osteoporosis with high bone fragility only in the osteoblast cKO. The effect was gender- and age-independent and independent of LRP5 [211] (Table 3).

### 3.2. WNT3A

WNT3A can signal through both the canonical and non-canonical Wnt pathways [18,19,20,21,22]. Although WNT3A is not expressed in osteoblasts, it can activate the canonical pathway in them to promote their differentiation (in vitro) [245,246]. Because of this feature, Wnt3a is, together with Wnt1, the most widely used Wnt ligand for the stimulation of canonical Wnt signaling, which induces cell proliferation and survival in osteoblasts [247,248]. In uncommitted osteoblast precursors and differentiated osteoblasts, Wnt3a also prevents apoptosis [225,245,249]. Wnt3a treatment does trigger a negative feedback mechanism, as demonstrated by the increase in Wnt antagonists (axin2, DKK2, SFRP2) and the decrease in WNT2B. 

In vitro, WNT3A plays an important role in the differentiation of MSCs into osteoblasts by the activation of the canonical pathway and the activation of PKC δ through the non-canonical pathway [18,250]. The expression of alkaline phosphatase (ALP), bone sialoprotein, osteocalcin and osterix is increased by WNT3A [18,248]. It is likely that, in vivo, the WNT3A concentration determines the fate of MSCs towards an adipogenic or osteogenic lineage through the activation of the expression of ALP and other bone markers. These results contradict those of Boland et al. that stated that WNT3A promoted the proliferation of osteoblast precursors in their undifferentiated state by the inhibition of ALP and bone sialoprotein [245,251]. Liu et al. demonstrated that, although WNT3A can inhibit osteogenic differentiation, inhibition towards the adipogenic lineage will prevail over the inhibition of osteogenic differentiation under lower WNT3A concentrations, thus promoting osteogenic differentiation [252]. WNT3A also inhibits MSC differentiation into chondrocytes [21]. 

A *WNT3A* mutation has been described in homozygosis in one patient with a skeletal dysplasia consisting of osteopenia, bilateral coxa valga deformity, mild left radial and ulnar bowing, broadening of metaphases and bilateral shortening of the great toes and thumbs [79] (Table 1, Figure 3). Wnt3a-KO mice were not viable due to several defects in embryonic development [213,214] (Table 3). 

### 3.3. WNT4

WNT4 plays a role in the differentiation of osteoblasts through the non-canonical pathways. While the expression levels of Wnt4 decrease during the proliferative expansion period of MSCs, they significantly increase during the osteoblastic differentiation stage [253]. In addition, BMP-induced osteogenic differentiation is enhanced in vitro in human and murine MSCs by WNT4 overexpression, which activates the non-canonical p38 MAPK-mediated signaling pathway without activating either the canonical or JNK pathway [254]. WNT4 also inhibits bone resorption and bone-loss-related inflammation [217]. WNT4 has also been reported to play a role in blood vessel development, suggesting that, in vivo, WNT4 may promote bone regeneration by promoting MSCs to form a suitable microenvironment to generate an entire bone/bone marrow structure [254].

In humans, the region containing *WNT4* and *ZBTB40* has been repeatedly reported to be associated with bone mass in different genome-wide association studies or SNP microarrays [137,141,255]. In vitro studies clearly showed that the knock-down of ZBTB40 in osteoblasts displayed disrupted osteoblast differentiation and mineralization while the WNT4 knock-down did not [216]. In vivo, Zbtb40-KO mice showed a reduction in lumbar spine areal BMD and bone volume, but not the decrease in femoral BMD observed in Wnt4-cKOs [216] (see below). Therefore, although a role of *WNT4* in bone cannot be ruled out, *ZBTB40* may also be responsible for the association of this region with bone mass.

Wnt4-KO mice die soon after birth and do not display any skeletal phenotype, although a delay in chondrocyte maturation has been observed [215,218] (Table 3). Wnt4-cKO in MSC showed decreased femoral BMD in females [216] (Table 3). The difference in results between the in vivo and in vitro KO models might reside in the paracrine effect of Wnt4 affecting bone resorption, which would not be observable in vitro [216]. On the other hand, overexpression of Wnt4 in differentiated osteoblasts leads to higher BMD and higher osteoblast number and activity [217] (Table 3). Estrogen-deficiency-induced and age-related bone loss were also lower in these Wnt4 overexpressing mice, with higher bone formation and lower bone resorption. Proinflammatory cytokines typical in estrogen deficiency bone loss were also inhibited under Wnt4 overexpression through the inhibition of the NFkB in macrophages and osteoclast precursors. Overexpression of Wnt4 partly mitigated the TNF inflammation-caused bone erosion phenotype in TNF + Wnt4 overexpressing mice [217]. Wnt4 suppresses RANKL-induced osteoclast differentiation by inhibiting the binding of the activated RANK to the TNF receptor-associated factor 6 (Traf6) in the osteoclast [217]. Mouse models have demonstrated that although the absence of Wnt4 can be compensated for in bone, its presence provides protection against bone resorption and strengthens bone formation. 

### 3.4. WNT5A

Although WNT5A has been historically classified as a non-canonical Wnt ligand, it has recently been shown that it can influence multiple Wnt pathways depending on the available FZD receptors [256]. Furthermore, recent studies have demonstrated that WNT5A can both activate and inhibit the canonical Wnt signaling pathway [205,247,256] and that it takes part in the differentiation of both osteoblasts and osteoclasts [103,221,257,258]. WNT5A is expressed in MSCs, osteoblasts and osteoclasts [257]. In vitro studies showed that in the absence of FZD4 and LRP5, cells responded to WNT5A by inhibiting the WNT3A-mediated canonical pathway [205]. WNT5A can induce β-catenin degradation independently of GSK3 phosphorylation [220] and through the non-canonical WNT/Ca^2+^ pathway, thus inhibiting TCF-mediated transcription downstream of β-catenin stabilization [256,259]. In the presence of FZD4 and LRP5 receptors, however, WNT5A can activate canonical Wnt signaling [205]. It has been suggested that sphingosine-1-phosphate (S1P) promotes the expression of WNT5A and LRP5 in osteoblasts [260], and that WNT5A promotes osteoblast differentiation via LRP5/6 expression in an osterix-dependent manner [221]. Similar to WNT10B, WNT5A inhibits stem cell differentiation into adipocytes, thus promoting osteoblastogenesis through the non-canonical pathways [261]. Wnt5a treatment stimulated bone nodule formation and ALP expression in hMSCs [257,262]. WNT5A plays an important role in osteoclastogenesis, especially during embryonic development. Osteoblast-lineage cells and osteoclast precursors express WNT5A and ROR2, respectively, and ROR2 signaling in osteoclast precursors enhances RANKL-induced osteoclastogenesis through the JNK/cJun/Sp1 pathway [103,258].

In humans, LoF mutations in *WNT5A* result in the dominant Robinow syndrome [60,247] (Table 1 and Figure 3). On the other hand, WNT5A is highly expressed in synovial tissues in rheumatoid arthritis (RA) patients. Excess WNT5A-ROR2 signaling may contribute to bone loss in this disease, as shown in rheumatoid arthritis models, where the inhibition of Ror2 signaling suppressed bone loss [103,258].

Wnt5a-KO mouse models show perinatal mortality and a variety of developmental aberrations, partly due to a delay in chondrocyte hypertrophy and skeletal ossification [103,219,263] (Table 3). Osteoblast lineage cells extracted from these mice displayed impaired mineralization and differentiation [221]. These embryos express increased β-catenin signaling in the distal limb, and *ex vivo* inhibition of the canonical pathway partially recovers the limb phenotype [219,220]. Heterozygous Wnt5a-KO mice are viable, although they show reduced BMD as a result of lower osteoblast and osteoclast number and increased adipogenesis [103]. Wnt5a-cKO in osteoblasts also resulted in impaired bone formation, but also in impaired osteoclast formation and bone resorption as they failed to upregulate RANK expression in osteoclast precursors [103] (Table 3). In mice, endogenous Wnt5a and Wnt5b regulate endochondral skeletal development by coordinating chondrocyte proliferation [220,263]. The Wnt5a-cKO in chondrocytes showed delayed osteoblast differentiation, suggesting that Wnt5a secreted from chondrocytes regulates osteoblast differentiation separately from the chondrocyte differentiation also necessary for bone formation [263] (Table 3). General overexpression of Wnt5a prior to E7.5 has lethal consequences. After this time point, multiple phenotypes can be obtained due to ectopic Wnt5a expression. Some of these phenotypes can be linked to the inhibition of canonical Wnt signaling and others to its activation. Referring to bone phenotype, Wnt5a overexpression causes delayed bone formation in the developing skull and limbs, suggesting impairment in both endochondral and intramembranous ossification processes [205]. However, when Wnt5a was overexpressed only in osteoblasts, no differences were detected compared with the wild type [225] (Table 3).

### 3.5. WNT5B

WNT5B is linked to bone development, especially to MSC differentiation and chondrocyte proliferation [263,264]. WNT5A and WNT5B play redundant roles in the differentiation of mesodermal progenitor cells into MSCs in vitro, and the inhibition of both ligands is necessary to impair this process. This differentiation is activated by the WNT5/calmodulin pathway [264].

*WNT5B* has been repeatedly associated with bone mass in GWAS studies [135,136,137,140,141,142]. Furthermore, *WNT5B* has been found to be upregulated in female osteoarthritis (OA) patients and in vitro in osteoblasts exposed to glucocorticoids [265,266].

In mice, Wnt5b, in coordination with Wnt5a, plays an important role in longitudinal bone growth by regulating chondrocyte proliferation. Both Wnt5b-KO and mice overexpressing Wnt5b in chondrocytes showed delayed chondrocyte differentiation. The latter resulted in delayed bone ossification [263], while the skeletal phenotype of Wnt5b-KO mice has yet to be described [222] (Table 3). 

### 3.6. WNT7A

WNT7A can act through the canonical and non-canonical Wnt pathways and plays an important role in limb and craniofacial development during embryogenesis [267,268].

In humans, complete and partial LoF mutations have been described in Al-Awadi-Raas-Rothschild syndrome (AARRS) and in the less severe Fuhrmann syndrome, respectively [75,267]. It is also responsible for the recently described Santos syndrome [76] (Table 1 and Figure 3). These syndromes share a phenotype of ectodermal dysplasia, with severe limb defects, among other characteristics [76,267].

Wnt7a-KO mice present impaired limb development, especially affecting digit number and formation [223] (Table 3). The polysyndactyly present in mice KO for En1, a Wnt7a suppressor, was restored in En1 + Wnt7a-dKO mice. Similarly, a reduction in Dkk1 expression, typically associated with polysyndactyly, resulted in the restoration of digit number in Dkk1db/- + Wnt7a-KO and Dkk1db/db + Wnt7a-KO mouse models, whereas other phenotypic characteristics were not restored. These results indicate that both Wnt7a and Dkk1 seem to cooperate to determine digit number [223].

### 3.7. WNT7B

WNT7B stimulates bone formation through non-canonical activation of mTORC1 and PKC δ [18,225]. In the developing long bones of the mouse, Wnt7b is enriched in the osteogenic perichondrium flanking the hypertrophic cartilage, but its expression declines in the more mature osteoblasts, indicating transient upregulation of Wnt7b during osteoblast differentiation in vivo, regulated by Ihh [18,210,269].

In humans, rare mutations in *WNT7B* have not been associated with any particular disease. However, some GWAS have reported the association of SNPs in the *WNT7B* region with heel BMD [140,141,142]. Additionally, increased WNT7B expression has been reported in the articular cartilage of OA patients [270,271] and in the macrophage-like cells, fibroblastic cells and vessel walls in the synovium in RA patients, particularly in regions of inflammation [270]. 

Wnt7b is a potent bone anabolic protein both during embryogenesis and in the postnatal life of mice. Specifically, it markedly increases both the number and function of osteoblasts [225]. Although total Wnt7b-KO results in perinatal death [224], deletion of Wnt7b in the skeletal osteoprogenitors causes only a modest delay in ossification that is resolved after birth [18,225] (Table 3). This delay is apparently connected to a delay in chondrocyte maturation and a deficit in osterix-mediated osteoblast differentiation [18]. On the contrary, overexpression of Wnt7b in preosteoblasts, mature osteoblasts or both osteoblasts and chondrocytes resulted in accelerated bone formation and denser bones in mouse models, with little bone marrow space due to its ossification [210,225] (Table 3). This phenotype alleviates with age, probably due to higher bone resorption [225]. Mice overexpressing Wnt7b in mature osteoblasts showed an increase in the amount of osteocalcin and osteoblast number, while the number of osteocytes was similar to the control. Conditional deletion of Raptor in mice overexpressing Wnt7b in preosteoblasts reduced the Wnt7b-induced hyperactivity of osteoblasts, but not their number [225]. Postnatal induction of Wnt7b in mice also resulted in increased BMD [225]. When Wnt7b was activated in aged mice, the increase in bone formation was higher than the increase in bone resorption, leading to denser bones with increased mineralization. Wnt7b also takes part in the later stages of bone healing, accelerating healing by means of increased mineralization [226]. Therefore, Wnt7b seems to play a role in bone formation but not to the same extent as in bone resorption. Wnt7b expression is minimal in the bones of young adult mice, but can be upregulated by mechanical bone loading [244]. This induction decreases with age in both mice and humans [226,244,272]. 

### 3.8. WNT9A

The main role of WNT9A in bone is related to chondrocyte regulation and is associated with the canonical Wnt pathway [273,274]. WNT9A is required for appropriate chondrocyte arrangement prior to endochondral ossification [274], and to maintain joint integrity [218]. WNT9A expression decreases in MSCs upon osteoblastic differentiation [245]. 

Wnt9a-KO mice die at birth and display skeletal abnormalities such as partial joint fusion, shortened long bones, reduced mineralization and chondroid metaplasia [218] (Table 3). The reduction in mineralization in null mice can be explained as a delay in chondrocyte maturation and therefore endochondral ossification. In addition, as signals from the joints are suspected to define the size of the skeletal elements, this would explain why a decrease in Wnt9a in the joints is associated with shorter long bones, through the spatial and temporal modulation of Ihh signaling during embryonic development. The phenotypes associated with synovial joints, as well as the delay in chondrocyte maturation, are augmented in Wnt9a + Wnt4 dKO [218] (Table 3). It is possible that the Wnt9a-mediated canonical pathway and the Ror pathway converge to regulate chondrocyte maturation. Ror2 + Wnt9a-dKO mice showed a delay in both chondrocyte and osteoblast maturation [166] (Table 3).

### 3.9. WNT10A

WNT10A plays an early role in osteoblastogenesis, favoring bone formation mainly through the inhibition of adipogenesis. Similar to WNT6 and WNT10B, WNT10A suppresses the differentiation of MSCs into adipocytes through the canonical Wnt pathway [258,275]. Because of this role, WNT10A is upregulated in the proliferative period of MSCs and its expression decreases upon differentiation into osteoblasts [253]. In vitro, demethylation of *WNT10A* has been seen to inhibit early adipocytic differentiation through the canonical pathway in both MSCs and preadipocytes [276]. 

Mutations in *WNT10A* are associated with ectodermal dysplasias, such as odonto-onycho-dermal dysplasia [277], as well as tooth agenesis due to the important role of WNT10A in odontoblast differentiation [78,278,279] (Table 1 and Figure 3). 

Wnt10a-KO mice show a decrease in bone mineralization and impaired bone and dental formation [227,228,229] (Table 3). Bone volume, mineral apposition rate and bone formation rate were reduced, while osteoclast and osteoblast number were similar to those in controls. Interestingly, these mice also showed a decrease in adipocytic differentiation and a decrease in adipose tissue in the bone marrow [227]. A Wnt10a-cKO in epithelial cells presented dental defects among other features, although the BMD was not altered [280] (Table 3). 

### 3.10. WNT10B

WNT10B activates the canonical pathway and plays an important role in osteoblastogenesis, inhibiting the expression of adipogenic transcription factors (C/EBPα and PPARγ) in MSCs while promoting osteoblastogenic transcription factors (RUNX2, DLX5 and osterix) [230,281]. Although WNT10B does not seem to play such an important role in bone resorption [231,232], osteoclast medium containing Wnt10b promotes osteoblastogenesis [282]. WNT10B is important in the maintenance and/or renewal of osteoblast progenitor cells [232]. There is evidence about the possible upregulation of Wnt10b expression under mechanical loading of the bone [283,284], but only under strong stimulation [244]. Wnt10b is expressed in the bone marrow, postnatal growth plate, osteoblastic precursors and various other stem cell compartments [232].

*WNT10B* has been implicated in several human diseases [285]. Homozygous missense mutations in *WNT10B* were demonstrated to cause split hand/foot malformation with autosomal recessive inheritance [72,247]. Mutations in *WNT10B* are also implicated in dental anomalies, including oligodontia [66,286] (Table 1 and Figure 3). In addition, it has been shown that the expression of *WNT10B* correlates with the survival rate in patients with osteosarcoma [287]. 

Studies in KO and transgenic mice have found that Wnt10b facilitates osteogenesis and increases bone mass [230]. Wnt10b-KO mice have a 25–30% decrease in bone volume fraction, BMD and trabecular number compared with WT controls [230] (Table 3). A reduction in the number and function of osteoblasts was detected in homozygous and heterozygous Wnt10b-KO mice. Interestingly, Wnt10b-KO animals showed an enhanced trabecular structure at 2–4 weeks of age that was rapidly followed by progressive osteopenia, suggesting that Wnt10b helps to maintain osteoblast precursors in an undifferentiated state and that, in its absence, the accelerated differentiation of mesenchymal osteoprogenitors occurs, depleting the stem cell pool [232]. Both osteogenesis and adipogenic centers were affected in this model. Both copies of Wnt10b are required for normal maintenance of adult bone homeostasis and the loss of bone in aged heterozygous animals is as severe as that for homozygous Wnt10b-KO mice [232]. Related to this, Wnt10b levels are downregulated in the absence of estrogen in ovariectomized animals, another model of osteoporosis [288]. On the other hand, transgenic overexpression of Wnt10b in mature osteoblasts and bone-marrow-derived mesenchymal precursor cells results in increased bone mass and bone strength [231] (Table 3). Mice overexpressing Wnt10b in MSCs show increased bone mass and are resistant to age-related and hormone-related bone loss (Table 3). Furthermore, these mice are genetically lean, have reduced adiposity and are resistant to obesity [230]. The increased bone mass in mice overexpressing Wnt10b in osteoclasts is caused by increased bone formation due to an elevated osteoblast number, while the adipose tissue of the mice is not affected (Table 3). The increase in mandibular bone results in delayed incisor tooth development [230,231].

### 3.11. WNT16

WNT16 is an important Wnt ligand involved in the regulation of postnatal bone homeostasis. The precise mechanism by which WNT16 acts on the different signaling pathways of WNT has not been fully identified. Different studies postulate that the effect of WNT16 on these pathways is cell-type-specific. While there are studies showing that it activates the canonical pathway in MC3T3-E1 preosteoblasts and in Saos2 osteoblast-like cells, WNT16 does not activate this pathway in HEK293 cells [234,289,290]. There is evidence that, in chondrocytes, MC3T3-E1 and perivascular stem/stromal cells, WNT16 is also capable of acting through non-canonical pathways (PCP/JNK [291] and JNK [234,292]). Altogether, these results indicate that WNT16 regulates both canonical and non-canonical Wnt targets in osteoblasts [293].

In humans, the genomic region of *WNT16* is one of the most consistent GWAS signals, and has been associated with different skeletal phenotypes, including BMD, bone strength, geometric parameters, cortical bone thickness and fracture risk [135,136,137,138,139,141,142,143,235,236,241,294,295,296,297,298]. In addition, WNT16 has been diversely associated with OA. The expression of WNT16 was found to be dramatically upregulated after cartilage injury or in OA onset [299]. Wnt16-deficient mice developed severe OA because of reduced lubricin and increased chondrocyte apoptosis [300]. Using chondrocyte-cKO animal models and overexpression of Wnt16 by intra-articular injection of adenoviral Wnt16 vectors, Tong et al. determined that Wnt16 inhibited chondrocyte hypertrophy in both skeletal development and OA pathology, therefore producing positive effects on cartilage and ongoing OA [291]. Contrary to these experiments, Törnqvist et al. [301] found that overexpression of Wnt16 in osteoblasts increased the subchondral bone mass but had no impact on OA in young adult female mice. 

Wnt16-KO and the osteoblast-specific Wnt16-cKO mouse models show spontaneous fractures as a result of low BMD, low cortical thickness, reduced bone strength and high cortical porosity, while the trabecular bone volume is not altered [233,234,235,236]. Consistent with this, Wnt16 was found to be more highly expressed in cortical bone than trabecular bone [234]. The phenotype of the osteocyte Wnt16-cKO mouse model is modest compared with the early osteoblast Wnt16-cKO [234]. On the other hand, when Wnt16 is overexpressed in osteoblasts and osteocytes, the mouse models show higher BMD and better bone resistance in both trabecular and cortical bone [238,239,240] (Table 3). The difference in effect in the trabecular bone phenotype between the KO and KI mice might be due to the low expression of Wnt16 in the trabecular compartment. Despite the obvious bone phenotype that Wnt16 produces in mouse models, the molecular mechanisms by which Wnt16 regulates bone metabolism are not yet fully understood. Some authors suggest that Wnt16 acts by inhibiting osteoclastogenesis both indirectly, through the increase in OPG, and directly, acting on the osteoclast progenitors by osteoblast-derived Wnt16 [234,302]. Other studies have shown that Wnt16 also exerts an effect on bone formation, increasing both the activity and the number of osteoblasts [233]. Different effects have also been observed in transgenic mouse models overexpressing Wnt16 in osteoblast cells; some do not show differences in bone formation and resorption [238], while others show a large increase in bone formation and a moderate decrease in bone resorption [239,240]. Moreover, Shen et al. demonstrated that Wnt16 promotes the in vitro osteogenic differentiation of human perivascular stem/stromal cells [292]. Taking all these studies into account, it would appear that Wnt16 has a dual anabolic and anti-resorptive effect. Different animal models show that the effects of Wnt16 and estrogen are independent [238], although it has been shown that the expression of Wnt16 in bone varies according to estrogen status [238,303]. Moreover, Wnt16 is a crucial regulator of cortical bone in young adults and old mice [237]. These discoveries make WNT16 a good candidate as a therapeutic target for the treatment of osteoporosis in postmenopausal women. Another interesting potential application of Wnt16 treatment would be in bone loss due to glucocorticoid treatment. In reference to this, studies in animal models with Wnt16 overexpression and glucocorticoid treatment have been performed, with contradictory results. While some report partial protection against glucocorticoid-induced bone loss [304], others report that Wnt16 overexpression does not prevent it [305]. Therefore, more studies will be needed to clarify whether WNT16 could be a good therapeutic solution.

## 4. Inhibitors

We can group the Wnt inhibitors into two categories depending on the action they perform: (i) interfering with LRP binding to FZD-WNT, such as the DKK protein family and sclerostin, and (ii) binding to the WNT ligands and preventing their binding to the membrane receptors, such as the SFRPs 1–5 and WIF-1. We review the role of these inhibitors, focusing on mutations in Human gene mutation database (HGMD) associated with diseases (Figure 4) and the different mice models available (Table 4).

### 4.1. DKK

In vertebrates, the Dickkopf (Dkk) family comprises four genes encoding secreted glycoproteins DKK1–4. Members DKK1, DKK2 and DKK4 have a similar structure, with two cysteine-rich domains (CRD 1/2) (Figure 4), and act through the canonical Wnt pathway, while DKK3 is the most divergent member of the family, with the addition of a soggy domain at the N-terminus and no effect through the canonical Wnt pathway [336,337]. 

DKK1 is by far the most studied member of the family due to its crucial role in head morphogenesis [27,338], specifically in bone tissue, since the the GoF mutations in *LRP5* causing HBM actually impair DKK1 binding, thus producing a more active Wnt pathway [39,132]. DKK1 performs its function by inhibiting the canonical Wnt pathway through binding with high affinity to LRP5/6 co-receptors. While it has been shown that DKK1, DKK2 and DKK4 can interact with LRP5/6, DKK3 does not [27,338,339,340,341,342]. The CRD2 domain binds, with high affinity, to the first and third β-propeller domains of the LRP5/6 co-receptors and to KRM1/2, and the NXI domain binds, with moderate affinity, to the first β-propeller domains of the LRP5/6 co-receptors [127,203,339,342,343,344,345]. As explained in Section 2.3, KRM and DKK1 binding facilitates DKK1 inhibitory activity by forming a heterotrimeric LRP5/6-DKK1-KRM1/2 complex that induces the rapid internalization of LRP5/6, decreasing the availability of these receptors in the cell membrane [168,169]. However, it can also have an inhibitory effect in the absence of KRM1/2 [168]. DKK1 is detected at low levels in most human tissues except the placenta [346,347], but, in a body-wide screen of Dkk1 expression in young and adult mice, strong expression of Dkk1 was found to be mainly restricted to bone [312]. This expression is most abundant in bone from young mice and decreases dramatically in adolescent and adult mice [348,349]. Dkk1 expression has been demonstrated in osteoprogenitors, osteoblasts, osteocytes and adipocytes [127,307,350,351], but there is controversy over whether the main source is early osteoblasts or mature osteocytes [310,352,353].

DKK1 has been implicated in the pathogenesis of various diseases affecting bone and is a crucial modulator of glucocorticoid-induced [265,354,355] and estrogen-deficiency-induced bone loss [355,356]. Deregulation of DKK1 expression levels is associated with various bone-related disorders, such as osteonecrosis of femoral head (ONFH) [357] and Paget disease; diabetes mellitus; autoimmune diseases such as RA, OA, ankylosing spondylitis and systemic lupus erythematosus; and various types of cancer, including multiple myeloma, breast cancer and prostate cancer [358,359]. Very little data exist on the potential role of DKK1 in bone mass in humans. To date, no disease-causing mutation has been reported in the HGMD. Recently, we described two mutations (p.Try74Phe and p.Arg120Leu) in patients with the HBM phenotype, both of which showed a loss of DKK1 inhibitory activity towards the Wnt pathway [55,56]. Nevertheless, the p.Arg120Leu mutation has also been found in patients with other phenotypes such as osteoporosis and anal malformations [360,361]. No SNPs have been found to be associated with BMD or with different bone parameters in several GWAS. However, many SNPs located in a region 350 kb downstream of DKK1 have been found to be associated with BMD [135,136,137,138,139,140,141,142,294,298]. Interestingly, a DKK1-activating long non-coding RNA (*LNCAROD*) maps in this region [362]. A recent study by our group confirmed the physical interaction between this region enriched in GWAS signals and the DKK1 promoter, using the circularized chromatin conformation capture (4C) assay [56].

Dkk1-KO mice die shortly after birth and exhibit developmental defects, including head defects and limb dysmorphogenesis [306] (Table 4). In contrast, the heterozygous Dkk1-KO mouse, the hypomorphic (doubleridge, db) mouse, the tamoxifen-inducible mouse and the osteolineage-specific mouse models show an increase in BMD and bone formation [307,308,309,310,311] (Table 4). By studying the allelic combination of null or db mice (Dkk1+/db; Dkk1+/−; Dkk1db/db; Dkk1db/−; Dkk1−/−), it was seen that the increase in BMD correlates dose-dependently with decreased expression of Dkk1 [308,309]. In contrast, transgenic Dkk1 overexpression causes osteopenia, with a decrease in the number of osteoblasts and in bone formation (Table 4) [312,313]. In dKO mouse models, it was shown that the developmental lethality and head morphogenesis defects of the Dkk1-KO were rescued by reduced Lrp6 or Wnt3 expression (Table 4) [309,316]. While all models share an effect on bone formation, some of the generated mouse models show no difference in bone resorption, whereas osteoblast lineage cell-specific or tamoxifen-inducible cKO mice show a significantly reduced number of osteoclasts and a reduced RANKL/OPG ratio [307,308,309,310]. In addition to these bone phenotypes, it was found that the response to dynamic tibial loading was augmented in mice lacking Dkk1, but that Dkk1 does not seem to be essential for transducing the response to cyclic loading [244,315,363].

Thanks to in vitro and in vivo experiments, DKK1 has been confirmed to affect osteoblast function and has been linked to inhibition of the differentiation of osteoblasts and chondrocytes. Moreover, DKK1 also provokes the stimulation of adipogenesis and the activation of osteoclasts, increasing the RANKL/OPG ratio [307,310,312,350,358].

While DKK1 acts as a pure inhibitor of the canonical Wnt pathway, DKK2 can inhibit or activate this pathway depending on the cellular context [127,168,169,364,365]. DKK2, as with DKK1, binds with high affinity to KRM1/2 and LRP5/6 through the CRD2 [127,169,364,365]. In vitro studies in HEK293T cells have revealed that, in the presence of KRM2, or low levels of LRP5/6, DKK2 behaves similarly to an LRP6 antagonist, preventing the binding of Wnt to the membrane co-receptor. In contrast, in the absence of KRM2 or the presence of high levels of LRP5/6, DKK2 can activate the LRP6 co-receptor independently of DVL [169,364]. The same occurs in the absence of WNT7b, where DKK2 inhibits osteogenesis in culture, and the presence of high WNT7b levels induces terminal osteoblast differentiation [317]. Although the homology between DKK1 and DKK2 is very high, especially in the CRD2 domain, there are some differences in affinity for the LRP5/6 transmembrane co-receptors. While DKK1 has higher affinity for LRP6, DKK2 has higher affinity for LRP5 [168,340]. Furthermore, the pattern of expression in osteoblasts and the regulation of gene expression are very different. While DKK2 expression is higher in osteoblasts than in osteocytes, that of DKK1 is higher in osteocytes [317].

The Dkk2-KO mouse model shows osteopenia, a decreased rate of bone formation and an increased number of osteoclasts due to the increased RANKL/OPG ratio (Table 4). Through the study of the differentiation of cultures of calvary osteoblasts, it was postulated that Dkk2 could affect the terminal differentiation of osteoblasts that leads to the formation of mineralized matrices [317].

### 4.2. Sclerostin and SOSTdc1

Sclerostin (encoded by the SOST gene) and SOSTdc1 (SOST domain-containing protein 1; also known as SOST1, USAG-1, WISE or ECTODIN) are paralogous secreted proteins and, due to their cysteine knot motifs, are classified into the DAN family of BMP antagonists [366,367,368]. Sclerostin and SOSTdc1 share 55% homology, with a very similar C-terminal domain. Both have been described as inhibitors of the BMP pathway and as WNT antagonists [325,369,370,371,372,373,374].

Sclerostin is a cysteine-rich glycoprotein that acts as a potent antagonist of the Wnt pathway, exerting its effect by blocking LRP5/6 co-receptors [369,370]. To perform this function, sclerostin binds to LRP4, thereby enhancing its suppressive effect on the canonical Wnt pathway [47,99,100]. The binding of sclerostin to LRP5/6 occurs through the first β-propeller domain, so sclerostin can inhibit the function of the Wnt ligands that bind to this domain (e.g., WNT1) but not those that bind through the third β-propeller domain (e.g., WNT3A) [203]. This function is confirmed by mutations causing the LRP5-HBM phenotype, which show a loss of sclerostin affinity [128,130]. Due to its abundant expression in osteocytes, sclerostin was originally categorized as a protein exclusive to mature osteocytes [375]. Accordingly, the highest levels of sclerostin were found in osteocytes surrounded by mineralized bone and distant from active bone surfaces [376]. However, new studies have reported, through histological examination, that *SOST* transcripts are also found in the bone marrow, cartilage, kidney, liver, lung, heart and pancreas [377]. In relation to bone, in addition to osteocytes, sclerostin expression has been found in osteoblasts, osteoclasts, mineralized hypertrophic chondrocytes, odontoblasts and cementocytes [378]. *SOST* expression has also been found in the developing embryo, where it plays a role in limb patterning, which explains the cases of syndactyly among patients with sclerosteosis [379].

In humans, *SOST* mutations are associated with conditions characterized by excess bone formation: sclerosteosis, van Buchem disease, craniodiaphyseal dysplasia and the HBM phenotype (Table 1 and Figure 4). Indeed, the first evidence of the role of *SOST* in bone was the identification of LoF mutations responsible for sclerosteosis, a very severe autosomal recessive disease, caused by a lack of functional sclerostin [44,45,46,380]. Furthermore, a 52 kb deletion of a region downstream of the gene, necessary for the correct expression of *SOST*, has also been found in patients with van Buchem disease [49]. Van Buchem disease is less severe than sclerosteosis and the difference in severity lies in the fact that, in van Buchem patients, there is some residual expression of the *SOST* gene, whereas, in patients with sclerosteosis, there is none [381,382]. Moreover, craniodiaphyseal dysplasia is an extremely rare sclerosing bone dysplasia associated with missense mutations in the signal peptide of *SOST* that cause a decrease in its extracellular secretion [43]. Gregson et al. [54] and our group [383] also associated mutations in *SOST* with the HBM phenotype. In addition to these diseases, different GWAS studies find an association of *SOST* with BMD and the risk of fracture [135,136,137,139,140,141,142,384]. Furthermore, altered expression of the gene has been found under special or pathological conditions. For example, although no sclerostin protein expression has been found in articular cartilage in healthy joint tissue, many studies have reported sclerostin protein expression in articular chondrocytes isolated from osteoarthritic joints or in rheumatic joint diseases. Nevertheless, it is unclear whether sclerostin plays a protective role or whether it mediates disease pathogenesis [385]. Several groups have shown elevated serum sclerostin levels in postmenopausal women [386,387], with aging [388,389,390,391,392], in patients with excess glucocorticoids [393], in multiple myeloma patients correlating with osteoblast function and low survival [394], in prostate cancer and breast cancer [395,396] and in vascular calcifications [397,398].

The importance of the *SOST* gene in the determination of BMD has been demonstrated using animal models. Sost-KO mice show a phenotype of high BMD and bone strength with a high number of osteoblasts and increased bone formation that reproduce the phenotype observed in patients with sclerosteosis or van Buchem disease [318,319] (Table 4). In these mice, no changes were observed in the number of osteoclasts or in bone resorption markers, although an increase in OPG was observed [318]. In order to identify the cell-type-specific contributions to HBM of the Sost-KO, Yee et al. [320] generated different cKOs and observed that, despite the fact that osteocytes are the main source of sclerostin, the cKO that best phenocopies the HBM of the Sost-KO is the Sost-cKO in osteoprogenitor cells of the limb mesenchyme rather than the Sost-cKO in early-stage osteocytes (Table 4). In contrast, mice that overexpress this gene present a low bone mass, lower bone resistance, a low number of osteoblasts and reduced bone formation [321,322,323,324]. In addition, the mouse model with specific overexpression in osteocytes shows high levels of Rankl [321,322] (Table 4). All these data lead to the conclusion that sclerostin is a potent inhibitor of bone formation and that it may also influence the stimulation of bone resorption in some way.

The transcription of the *SOST* gene is finely regulated by a large number of signals. SOST expression is controlled by a proximal promoter region and by a 225 bp bone-specific distal enhancer called Evolutionary Conserved Region 5 (ECR5) 35 kb downstream of the gene that is physically interacting with the SOST proximal promoter [383,399,400]. The proximal promoter region (~1.4 kb upstream from the *SOST* gene) contains binding sites for RUNX2, osterix and SMAD2/3, among others, which promote *SOST* transcription [401,402,403,404,405,406]. In addition, the proximal promoter contains three estrogen response elements (ERES) that negatively control the expression of *SOST* through estrogen receptor B [407]. Furthermore, it has been determined that this region is subject to epigenetic modifications that regulate its expression, such as methylation (hypermethylated in osteoblasts to hypomethylated in osteocytes) [408,409] or acetylation (deacetylation of H3K9 by Sirt1) [410]. In addition to the proximal promoter, the ECR5 enhancer region regulates the correct expression of *SOST* in bone cells. This region is necessary and sufficient for the correct expression of the gene in osteocytes, although, to achieve the high levels of expression that occur, the proximal promoter is also required [400]. The ECR5 region is deleted in patients with van Buchem disease, and its deletion in mice causes a drastic decrease in *SOST* levels in osteocytes [400]. ECR5 stimulates the expression of *SOST*, in part through the binding of transcription factor Myocyte-enhancer factor 2c (Mef2c) [324,400]. The importance of the Mef2c transcription factor in the enhancing activity of ECR5 was confirmed with the Mef2c-cKO mouse in osteocytes/osteoblasts, which presented HBM and low levels of sclerostin [411]. Besides regulation through the proximal promoter, there are a large number of regulators of the expression of *SOST* in bone whose action is mediated via ECR5. The most relevant negative regulators of *SOST* are: PTH, PTH receptor, cyclic adenosine monophosphate (cAMP), protein kinase A (PKA), mechanical loading and the IL-6 family of cytokines. 

Sclerostin is the first molecular mediator described in osteocytes as responsible for the coupling of mechanical forces to the anabolic response of the skeleton [284,363]. The mechanical loading of bone results in a dramatic reduction in *SOST* expression resulting in increased bone formation, and, conversely, unloading increases it [363]. This is consistent with the fact that the Sost-KO mouse does not lose BMD after unloading [412], while the *SOST* transgenic overexpression mouse shows a reduction in load-induced bone formation [322]. The mechanical loading negatively regulates *SOST* expression by mechanisms that involve prostaglandin E2 production, nitric oxide and periostin expression [413,414,415,416]. While sclerostin is absolutely necessary for the anabolic actions of mechanical loading, it appears dispensable for PTH-induced bone gain [399,417]. Sclerostin is capable of uncoupling bone formation and bone resorption, inhibiting osteoblast function while stimulating osteoclast function. In addition to these effects, sclerostin has been shown to enhance adipocyte differentiation and promote adipogenesis and adipose hypertrophy via suppression of the Wnt pathway in in vitro and in vivo experiments [418,419,420].

Humanized monoclonal antibodies against sclerostin have been developed for the treatment of different bone pathologies, such as osteoporosis and OI [421,422,423,424,425,426]. Antibody treatment against sclerostin stimulates bone formation, decreases bone resorption and increases bone mass and strength while decreasing the risk of fragility fracture in both animals and humans [427,428,429]. Thanks to positive results in clinical phase 2 and 3 trials for the treatment of postmenopausal osteoporosis [421,422,423,424], it is currently approved for the market. The negative aspects associated with anti-sclerostin antibody treatment are that the increase in bone formation is not sustained in the long term and that it is contraindicated in patients with hypocalcemia or a high risk of ischemic heart disease or cerebrovascular disorder, therefore requiring caution before the implementation of treatment [430].

In the case of SOSTdc1, many studies, both in vivo and in vitro, have shown that it acts as an inhibitor of the Wnt pathway in a LRP5/6-dependent manner [325,368,372,431,432,433]. Despite this, studies in Xenopus embryos have shown that SOSTdc1 can activate or inhibit the Wnt pathway in a context-dependent manner [368]. Furthermore, SOSTdc1 can also bind to LRP4 [154,434]. Additionally, a mechanism has been described by which the retention of SOSTdc1 in the endoplasmic reticulum causes a decrease in LRP6 available on the cell surface [435]. Both sclerostin and Sostdc1 are widely distributed in the tissues of both the developing embryo and the adult mouse, with Sostdc1 being the most widely distributed. When expressed in the same organ system, they present non-overlapping expression patterns or different temporal phases, suggesting that these genes have evolved different sub-specializations within the signaling pathways they regulate, as a function of their cellular location [325,379]. For example, while Sostdc1 is expressed in mesenchymal preosteoblasts during the early period of bone formation (1.5–3 months), at 4 months, there is no expression of Sostdc1 in the osteoblast lineage, while sclerostin is expressed abundantly in both osteoblasts and osteocytes at this time. Hence, Sostdc1 and Sost are differentially expressed in osteoblast cells during temporally distinct phases, whereby Sostdc1 is expressed early and Sost later [325].

Sostdc1-KO mice showed a significant transient increase in BMD in the early stages (during the first 1.5 and 3 months), produced by an increase in the proliferation and number of osteoblasts (1.5 months), a decrease in the number of osteoclasts and an increase in bone formation rate (2.5 months) [325] (Table 4). In addition, there was dysfunction of the round proliferative chondrocytes, indicating that Sostdc1 possibly acts as an activator of the Wnt pathway in chondrocytes and as an inhibitor in the osteoblast lineage. A Sostdc1 spontaneous deficient mouse was shown to have severe tooth defects due to the loss of inhibition of the Wnt pathway [325,436,437,438]. However, another Sostdc1-KO model was found to lack any skeletal patterning defects or any obvious limb patterning defects [379] (Table 4). The Sostdc1 + Lrp5-dKO mouse shows normal BMD, thus indicating an interaction between Lrp5 and Sostdc1 [325]. Furthermore, the SOST + SOSTdc1-dKO mouse shows a 50% increase in the number of hand defects, suggesting partially redundant and complementary roles for these proteins in limb development [379]. The absence of sclerostin and Sostdc1 in limbs disrupts the epithelial–mesenchymal communication required for proper limb patterning in these dKO mice [379]. 

### 4.3. SFRPs

SFRPs are a family of secreted glycoproteins with a CRD structure in the N-terminus that shares 30–50% similarity with that of FZD receptors and a Netrin-related motif (NTR) in the C-terminus [371,439,440] (Figure 4). Thanks to their homology with FZD receptors, SFRPs can bind and sequester Wnt ligands [371] but also FZD receptors [441,442], blocking both canonical and non-canonical Wnt pathways. In the case of direct SFRP–Wnt ligand binding and inhibition, there seems to be a preference for certain combinations. For example, SFRP1 and SFRP2 are able to bind to WNT5A, while SFRP3 and SFRP4 cannot, and SFRP3 can inhibit WNT1 and WNT8 but not WNT3A or WNT5A [443]. This suggests a more complex and specific regulation of the pathway that might also involve the NTR domain and post-translational modifications [371,439,440,443,444]. There are conflicting results as to whether the CRD and NTR domains are sufficient on their own to perform Wnt binding and inhibition [445,446,447]. These conflicting results might imply that there is a biochemical/functional specificity in SFRP–Wnt interactions in different developmental models [371].

Despite being known as Wnt pathway inhibitors, SFRPs have also been found to activate the pathway. Low concentrations of SFRP1 can stimulate Wnt canonical pathway activation, while inhibiting it at higher concentrations in S2 cells in vitro [447]. In HEK and HSG cells, overexpression of Wnt3a and SFRP2 activated the canonical pathway more than Wnt3a overexpression alone [448]. In the case of bone cells, the combination of Wnt3a treatment with SFRP1 or SFRP3 resulted in decreased cell proliferation but increased ALP activity and osteocalcin marker and mineral nodule formation in MSCs that was greater than any of the treatments alone. In this case, osteoblast differentiation was stimulated without the involvement of the canonical Wnt pathway [443,449]. Moreover, SFRPs can also antagonize one another’s activity [450], favoring Wnt signaling pathways. In addition, SFRPs can enhance the extracellular transport of Wnt proteins [451], modulate BMP signaling [452] and interact with other receptors or matrix molecules [453]. It was hypothesized that SFRPs may have concentration-dependent, tissue-specific and/or stage-specific effects on MSC differentiation. This would result in small effects on undifferentiated mesenchymal stages that would increase in significance in pre-osteoblasts or mature osteoblasts in vitro [443]. Besides embryonic development, SFRP1–4 have been strongly linked to bone development, while SFRP5′s involvement in bone has barely been studied [440].

SFRP1 is an important regulator of osteoblast and osteocyte survival in vitro [327]. Expression of SFRP1 increases during osteoblast differentiation and peaks in the post-proliferative preosteocytic stage [327]. The higher expression of SFRP1 correlates with the inactivation of the canonical Wnt pathway and acceleration of osteoblast and osteocyte apoptosis in vitro and in vivo [327,328]. In vitro deletion of SFRP1 enhances osteoblast proliferation, differentiation and function, while suppressing osteoblast and osteocyte apoptosis [327]. In vivo, Sfrp1-KO mice have an increased bone trabecular area and are protected against age-related bone loss by the slowing down of osteoblast apoptosis and a modest enhancement of osteogenesis and mineralization (Table 4) [327,328]. Chondrogenesis and endochondral bone formation are also affected [454]. Moreover, Sfrp1-KO mice show an advantage in fracture healing due to early bone remodeling. Even though osteogenesis is favored over chondrogenesis, physiologic regulation of Wnt signaling is maintained, possibly thanks to SFRP redundancy, and osteoblast maturation continues as normal [454]. As expected, overexpression of Sfrp1 in mice resulted in lower bone formation and osteopenia (Table 4) [329]. Osteoblast activity markers (Propeptide type 1 N-terminal procollagen and OPG) were at similar levels compared to those of the wild type, but the markers for osteoblast maturation (OC) and bone renovation (RANKL) were lower. Cells derived from Sfrp1 Tg mice had lower ALP activity and mineralized nodule formation. Overexpression of SFRP1 did not affect bone resorption in vivo but induced higher osteoclastogenesis in vitro [327,329].

On the contrary, overexpression of SFRP2 in MSCs resulted in decreased apoptosis and increased MSC proliferation. Osteogenic and chondrogenic differentiation of the MSCs was inhibited by SFRP2 and could not be retrieved by the addition of recombinant Wnt3a or BMP2. Alfaro et al. proposed that SFRP2 might modulate the expression of Runx during osteogenesis via either the Wnt canonical or BMP pathways [452]. Similarly to SFRP1, SFRP2 negatively affects bone formation in vitro but the effects in vivo are relatively mild [12]. Sfrp2-KO mice showed defects in digit formation, due to decreased chondrocyte proliferation and delayed differentiation in distal limb skeletal elements [330] (Table 4). 

SFRPs show a level of redundancy during embryonic development, as single-gene SFRP-KOs showed mild phenotypes that were magnified in the case of double or triple KOs, resulting in embryonic lethality in most cases, such as for Sfrp1 + Sfrp2-dKO and Sfrp1 + Sfrp2 + Sfrp5 triple KO but not Sfrp1 + Sfrp5-dKO or Sfrp2 + Sfrp5-dKO mice [333,371]. Moreover, Sfrp1 + Sfrp2-dKO embryos exhibited craniofacial defects, limb outgrowths, extra digits and a shortened thoracic region [326,333]. 

SFRP3 expression is also related to the late osteoblast differentiation stage [327,455]. In MSCs, only SFRP3 promoted apoptosis [443,449]. Contrary to SFRP2, which facilitates infiltration and the metastasis of osteosarcomas [258], SFRP3 is downregulated in osteosarcoma patients at the local and systemic levels [456] and upregulated in bone marrow multiple myeloma samples with advanced lytic bone disease (LBD) as compared to no/limited LBD [457]. 

SFRP4 is highly expressed in skeletal regions during the late embryonic and postnatal developmental stages, in cells of both the osteoblastic and osteoclastic lineages. It is also strongly involved in postnatal skeletal development and aging-related and oxidative stress-related bone loss [331,332,458]. Mutations in *SFRP4* are linked to metaphyseal dysplasia or Pyle’s disease, characterized by a thinning of the cortical bone, limb deformity and bone fracture [61] (Table 1 and Figure 4). Elevated SFRP4 levels in tumors are associated with osteomalacia, whereby the bone becomes softer and more flexible [440,459]. SFRP4 is also highly expressed in chondrocytes from osteoarthritic patients, where there is a correlation between SFRP4 expression and chondrocyte apoptosis [460].

As SFRP4 alterations are responsible for Pyle’s disease, a Sfrp4-KO mouse was presented as a model for the disease [61]. However, as a single-gene KO, Sfrp4-KO mice showed increased trabecular bone mass and decreased cortical bone mass and thickness [61,331] (Table 4). This thinning was due to the activation of the Wnt5a/non-canonical pathway, which induces BMP2 and consequent *SOST* expression, leading to reduced bone formation [61,106]. Both Rankl and Opg were overexpressed in the cortical bone, with the balance favoring Rankl [61]. Osteocalcin and alkaline phosphatase were strongly expressed in Sfrp4-KO trabecular bone, in which mature osteoclast markers and activity were also decreased, whereas osteoblast markers were increased [331]. In aging Sfrp4-KO mice, trabecular and cortical bone density was maintained (compared to younger mice and aged control groups, respectively), showing that bone loss under aging and oxidative stress is related to the reactivation of sFRP4 gene expression [331]. In mice overexpressing Sfrp4 in osteoblasts, a decrease in trabecular bone thickness and volume was detected [332] (Table 4). In contrast to Sfrp4-deficient mice, bone resorption was not affected [332]. In this context, recent results reported by Chen et al. reinforce the importance of the non-canonical Wnt pathways in osteoclasts. Comparing Sfrp4-KO and the double KO mouse model Sfr4-KO + Ror2-cKO in osteoclasts, these authors determined that, although Sfrp4 inhibits both the canonical and non-canonical pathways, the increase in osteoclasts observed is due to the activation of the non-canonical pathway Ror2/Wnt/Jnk, overriding the activation of the canonical pathway that would be inhibiting bone resorption [106].

### 4.4. WIF-1

Wnt inhibitory factor-1 (WIF-1) is an evolutionarily conserved, secreted Wnt inhibitor. Even though its mechanism of action is similar to that of SFRPs, WIF-1 does not share sequence similarities with the CRD of the frizzled receptors. Instead, it is composed of five epidermal growth factor-like repeats and one WIF domain that mediates Wnt binding [461,462]. WIF-1 is a downstream target of the Wnt canonical pathway, suggesting that it may act as a negative feedback regulator. For instance, WIF-1 expression is increased during BMP-2 treatment of MSCs as well as during osteoblast differentiation [334,455,462]. Both WIF-1 and SFRP2 are induced in the late phase of osteoblast differentiation [455]. WIF-1 inhibits osteoblastic differentiation, ALP activity and mineralization and enhances adipogenesis. 

The most intriguing pathogenic alterations of this gene are related to its methylation pattern. A proposed treatment against osteoporosis, Gossypol, was observed to promote bone proliferation through the methylation and inhibition of the *WIF-1* promoter [463]. On the other hand, *WIF-1* methylation has been associated with bone tumors. *WIF-1* is epigenetically silenced in human osteosarcoma cell lines and most primary cultures [334]. 

Although Wif-1 is highly expressed in the developing and adult skeleton in vivo, it does not seem indispensable for its normal development, as Wif1-KO mice have no skeletal abnormalities (Table 4). This might be caused by functional redundancy in vivo [334]. Curiously, Wif1-KO mice are more susceptible to spontaneous and radiation-induced osteosarcomas, but not other cancer types [334]. Osteoblast-specific Wif1 overexpressing mice have, as with Wif1-KOs, no bone phenotype, but the overexpression of Wif-1 disrupts stem cell quiescence and self-renewal potency in the osteoblasts of these mice. Interestingly, this study found that overexpression of Wif-1 induced the autocrine induction of Wnt ligands in a compensatory way [335] (Table 4).

## 5. Future and Conclusions

The Wnt pathway is one of the most studied signaling pathways within the scientific community due to its involvement in different mechanisms, ranging from development and cancer to bone homeostasis. To date, numerous advances have been made in the knowledge of this pathway, but, due to its extreme complexity, there are still aspects to be identified, such as the differences in the performance in space (specific tissue) and time (from development to ageing). Specifically, for bone health, understanding these mechanisms is of great interest for the identification of therapeutic targets that allow the treatment of very common bone diseases such as osteoporosis or very severe conditions such as osteogenesis imperfecta. This is represented by the recent anti-sclerostin antibody treatment, which is providing very promising results in clinical management [421,422,423,424,425,426]. In the last year, other new advances in this topic have been published, including the positive effects of LRP5 (HBM mutation or overexpression) in conditions such as hyperglycemia and breast cancer, respectively [464,465], illustrating the relevance of crosstalk between bone and other tissues and the importance of studying the Wnt pathway at a physiological level beyond bone. Other areas of future interest should be the ageing bone, osteoarthritis and the relationship with vascular calcification. For this reason, a comprehensive study of the classic elements of the Wnt pathway (e.g., LRP5, DKK1, WNT1) as well as other less studied extracellular elements (e.g., KREMEN, SFRP4), which may be analyzed through the development and characterization of appropriate animal models, can provide us with new useful therapeutic targets.

## Figures and Tables

**Figure 1 genes-13-00138-f001:**
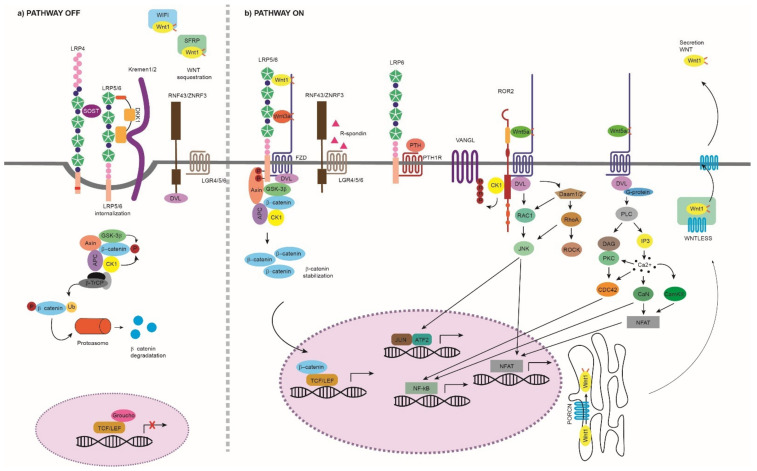
Wnt pathway. (**a**) Wnt pathway OFF: The Wnt pathway is inactive in the absence of Wnt ligands or by the effect of extracellular inhibitors that prevent the activation of the pathway. We can group the extracellular inhibitors into two categories depending on the action they perform: (i) binding to the Wnt ligands and preventing their binding to the membrane receptors, such as SFRP1–5 and WIF-1, and (ii) interfering with the LRP binding to FZD, such as DKK1 and sclerostin. Due to their mechanism of action, the first type of inhibitors can act on both canonical and non-canonical pathways, while the second type is specific to the canonical Wnt pathway. The inhibition performed by DKK1 and sclerostin is mediated or enhanced by receptors such as Kremen proteins and LRP4 [27,28]. When the canonical pathway is not activated, β-catenin is phosphorylated at three critical residues and sequestered in the cytoplasm by the destruction complex. The destruction complex is formed by scaffold proteins APC and AXIN1; Ser/Thr kinases, CK1α, ε and δ and GSK3β; and transcriptional regulators YAP/TAZ of the Hippo pathway [29]. This phosphorylated β-catenin is polyubiquitinated for its degradation in the proteasome by the complex containing, among others, β-TrCP protein. Thus, it is not available in the nucleus and, in its absence, a complex is formed by TCF/LEF and TLE/Groucho, inhibiting the transcription of target genes. (**b**) Wnt pathway ON: The binding of a Wnt ligand to FZD/LRP or ROR1/2/RYK/FZD activates the Wnt pathway that will result in a different transcriptional regulation. Canonical Wnt Pathway ON: The pathway begins at the cell surface with the formation of a heterotrimeric complex consisting of a Wnt ligand (19 different Wnts), a LRP transmembrane co-receptor (LRP5/6) and an FZD receptor (10 different FZDs). The formation of the LRP-FZD-Wnt complex results in phosphorylation of the LRP5/6 co-receptor by CK1α and GSK3β. Then, the DVL (also called DSH) polymerizes and is activated, inhibiting the destruction complex. This produces stabilization and accumulation in the cytoplasm of β-catenin, which will translocate to the nucleus. Once there, it displaces the TLE/Groucho repressors, forming an active complex with TCF/LEF proteins, which results in the recruitment of coactivators and activation of transcription of genes important for the differentiation and formation of bone, such as WISP1 and RUNX2 [12,29]. Non-canonical Wnt Pathway ON: (i) WNT/PCP: The WNT-PCP pathway begins with the binding of a Wnt ligand, such as WNT5A, to FZD and the co-receptors ROR1/2 or RYK. Then, DVL is recruited and activated, resulting in the activation of the scaffold protein VANGL. DVL forms a complex with DAMM1, which activates the small GTPases RHOA and RAC1, which in turn activate ROCK and JNK. This leads to rearrangements of the cytoskeleton and/or the induction of transcription through ATF2 and/or NFAT. The activation of the WNT5A-ROR1/2 pathway inhibits the canonical Wnt signaling [24,25]. (ii) WNT/Ca^2+^: The binding of the ligand to an FZD receptor results in the recruitment and activation of DVL. Then, DVL binds to the small GTPases, which activate PLC. This activation leads to the breakage of PIP2 into DAG and IP3. When IP3 binds to its receptor on the endoplasmic reticulum, calcium release occurs. When the calcium concentration is increased, DAG activates PKC, which in turn can activate CDC42. Increased intracellular calcium can also activate calcineurin and CaMKII, which in turn can induce the activation of the NFAT transcription factor or NF-kB [26,30]. WNT signaling modulators: The binding of RSPO to LGR and to RNF43/ZNRF3 maintains the Wnt signal ON by preventing the polyubiquitination and endocytosis of FZD performed by RNF43/ZNF3 [31]. WNT production: Wnt precursors undergo post-translational modifications such as porcupine-mediated palmitoylation, other lipid modifications and glycosylation in the ER. Then, the transmembrane protein Wntless (Wls) transports the functional Wnt ligands to the plasma membrane via the golgi apparatus. Wnt ligands are secreted from the cell by solubilization, exosome formation or on lipid protein particles [32,33,34]. See next figures for the information on the different protein domains. APC: adenomatous polyposis coli; AXIN1: axis inhibition protein 1; β-TrCP: β-Transducin repeat-containing protein; CAMKII: calcium/calmodulin-dependent protein kinase II; CDC42: cell division control protein 42; CK1: casein kinase 1; DAAM1: DVL-associated activator of morphogenesis; DVL: disheveled: FZD: frizzled; GSK3β: glycogen synthase kinase 3β; IP3, inositol 1,4,5 triphosphate, JNK: JUN kinase; LGR: leucine-rich repeat-containing G-protein-coupled receptor; NFAT: nuclear factor of activated T cells; NF-kB: nuclear factor kappa B; PKC: protein kinase C; PLC: phospholipase C; RAC: Ras-related C3 botulinum substrate; RHOA: Ras homolog gene family member A; ROCK: Rho kinase; ROR1/2: bind tyrosine kinase-like orphan receptor 1 or 2; RYK: receptor-like tyrosine kinase; PORCN: porcupine; RNF43/ZNRF3: ring finger protein 43/zinc and ring finger 3; RSPO; R-spondin ligand family members; SFRP: secreted frizzled-related proteins; TCF/LEF: T-cell factor/lymphoid enhancer factor; TLE: Transducin-Like Enhancer of Split Proteins; VANGL: Van Gogh-like; YAP/TAZ: Yes-associated protein/transcriptional co-activator with a PDZ-binding domain; WIF-1: Wnt inhibitory factor 1.

**Figure 2 genes-13-00138-f002:**
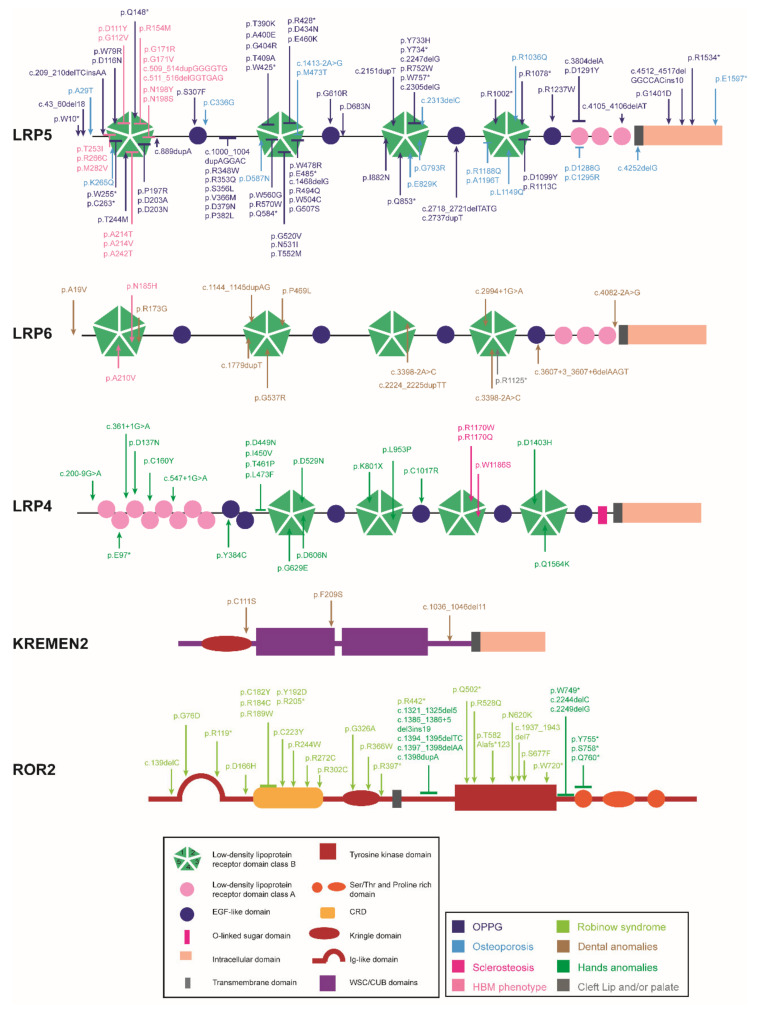
Domain structure of the co-receptors and localization of the mutations causing human skeletal diseases according to human gene mutation database (HGMD). ↓ indicates a point mutation, ⊥ encompasses more than one aminoacid position (a cluster of point mutations). *: STOP codon.

**Figure 3 genes-13-00138-f003:**
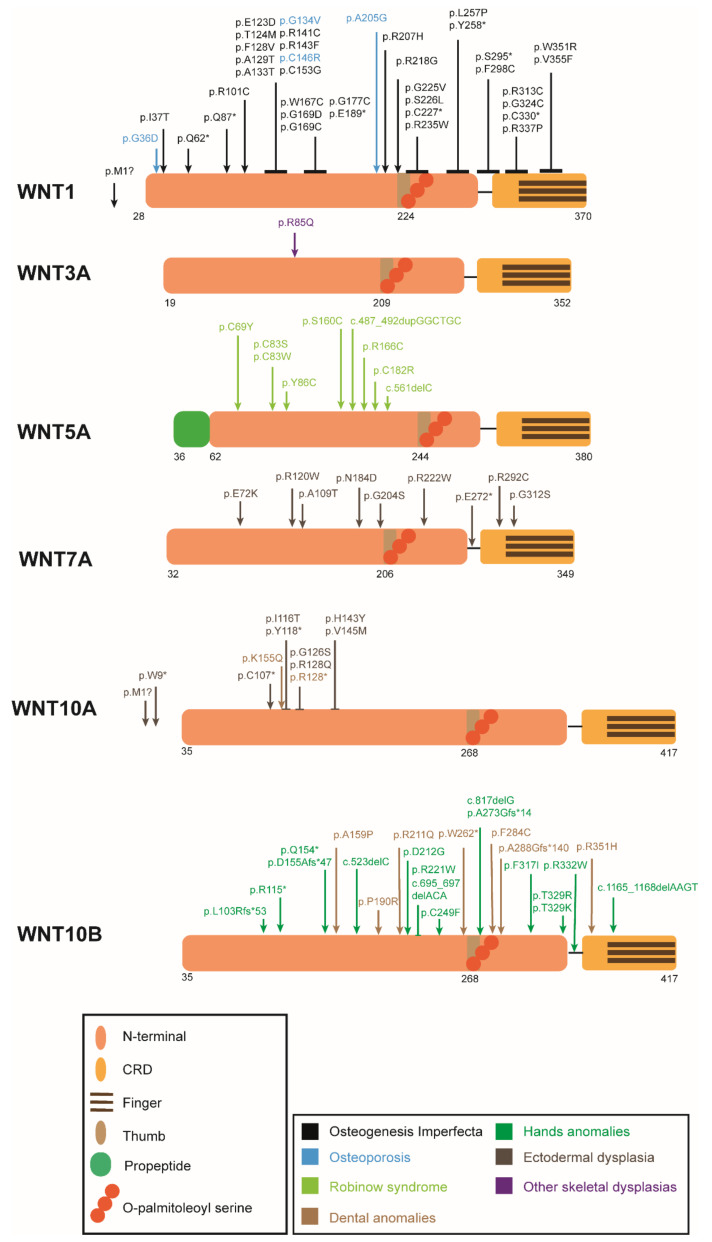
Schematic protein representation of the Wnt ligands with the mutations causing human skeletal diseases according to HGMD. ↓ indicates a point mutation, ⊥ encompasses more than one aminoacid position (a cluster of point mutations). Numbers below the structure show the amino acid position in the peptide before post-translational modification. *: STOP codon.

**Figure 4 genes-13-00138-f004:**
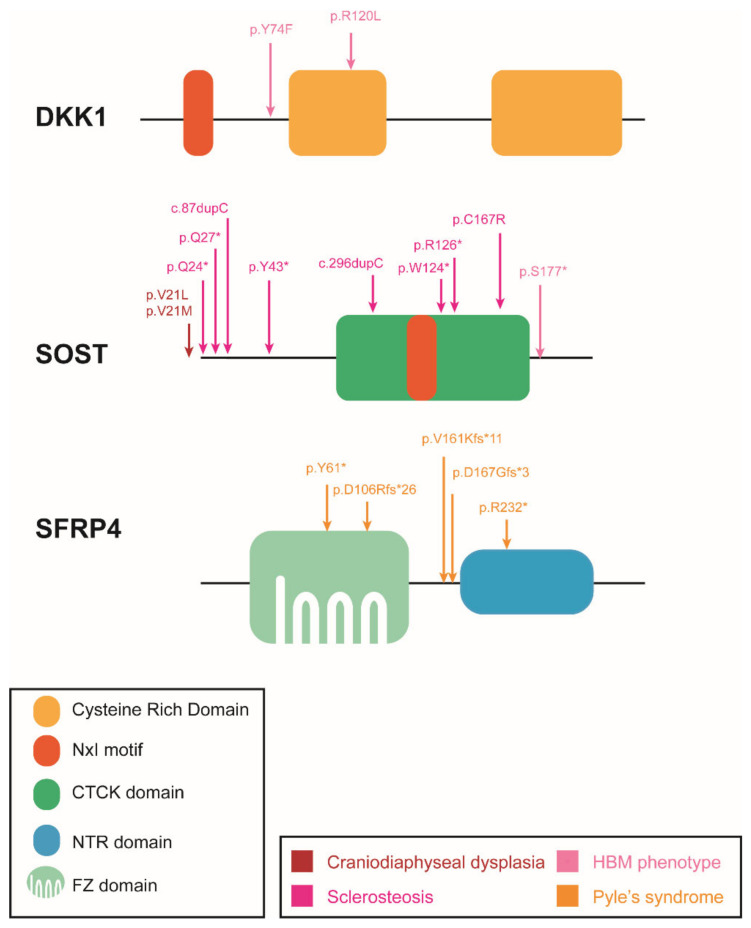
Schematic protein representation of the Wnt inhibitors with the mutations causing human skeletal diseases according HGMD. *: STOP codon.

**Table 1 genes-13-00138-t001:** Diseases or traits caused by mutations in Wnt pathway genes.

Disease/Trait	OMIM	Phenotype	Gene	Inh.	Comment	Refs.
Osteogenesis imperfecta XV	615220	- Low bone mass - Increased bone fragility	*WNT1*	AR/AD	LoF	[35,36,37]
Osteoporosis-pseudoglioma syndrome (OPPG)	259770	- Congenital blindness - Severe juvenile-onset osteoporosis - Spontaneous fractures	*LRP5*	AR	LoF	[38,39]
Osteoporosis	615221	- Low bone mass	*WNT1*	AD	LoF	[35]
	166710		*LRP5*	AD	LoF	[40,41,42]
Craniodiaphyseal dysplasia	122860	- Massive generalized hyperostosis - Sclerosis, especially involving the skull and facial bones	*SOST*	AD	LoF/DN	[43]
Sclerosteosis	269500	- Progressive skeletal overgrowth - Syndactyly	*SOST*	AR	LOF	[44,45,46]
	614305		*LRP4*	AD/AR	LOF	[47,48]
van Buchem disease	239100	- Hyperostosis of the skull, mandible, clavicles, ribs and diaphyses of the long bones, and tubular bones of the hands and feet.	*SOST **	AR	*** 52 kb downstream deletion	[49]
HBM phenotype ^1^	144750 607634 607636	- High bone mass, affecting especially the skull and tubular bones - Low fracture risk	*LRP5*	AD	GoF	[50,51,52]
			*LRP6*	AD	GoF	[53]
			*SOST*	AR	LoF	[54]
			*DKK1*	AD	LoF	[55,56]
Robinow syndrome	268310	- Dysmorphic facial features - Hypertelorism - Short-limbed dwarfism - Vertebral segmentation - Short stature, genital hypoplasia	*ROR2*	AR	LoF	[57,58,59]
	180700		*WNT5A*	AD	LoF	[60]
Pyle’s syndrome	265900	- Broadening of the long bones with wide and expanded trabecular metaphyses - Cortical thinning - Bone fragility and fractures - Genu valgum - Widening of the ribs and clavicles - Platyspondyly	*SFRP4*	AR	LoF	[61]
Selective tooth agenesis, types 7, 4 and 8	616724	- Absence of one or more teeth	*LRP6*	AD	LoF	[62,63]
	150400		*WNT10A*	AD/AR	LoF	[64,65]
	617073		*WNT10B*	AD	LoF	[66]
**Hand/foot anomalies:**						
Cenani–Lenz syndactyly syndrome	212780	- Complex syndactyly of the hands - Bone malformations in the forearm and lower extremities	*LRP4*	AR	LoF	[67]
Isolated bilateral syndactyly		- Fusion of two or more fingers	*LRP4*	AR	LoF	[68,69]
Brachydactyly type B1	113000	- Disproportionately short fingers	*ROR2*	AD	GoF	[70,71]
Split hand/foot malformation, isolated form, type 6	225300	- Syndactyly - Median clefts of the hands and feet - Aplasia and/or hypoplasia of the phalanges, metacarpals and metatarsals	*WNT10B*	AR	LoF	[72]
**Ectodermal dysplasia:**						
Ectodermal dysplasia 13	617392	- Severe oligodontia - Anomalies of hair and skin	*KRM1*	AR		[73,74]
Al-Awadi-Raas-Rothschild syndrome (AARRS)	276820	- Severe limbs malformations - Severely hypoplastic pelvis - Abnormal genitalia	*WNT7A*	AR	complete LOF	[75]
Furhman syndrome	228930	- Bowing of the femurs - Hypoplasia of the fibula, pelvis, fingers and fingernails - Cleft lip and palate	*WNT7A*	AR	partial LOF	[75]
Santos syndrome	613005	- Fibular agenesis or hypoplasia - Clubfeet with oligodactyly - Acromial dimples - Motion limitations of the forearms and/or hands - Severe nail hypoplasia or anonychia	*WNT7A*		LoF	[76]
Schöpf–Schulz–Passarge syndrome	224750	- Multiple eyelid apocrine hidrocystomas - Palmoplantar keratoderma - Hypotrichosis - Hypodontia - Nail dystrophy	*WNT10A*	AR	LoF	[77]
Odonto-onycho-dermal dysplasia (OODD)	257980	- Hyperkeratosis and hyperhidrosis of the palms and soles - Atrophic malar patches - Hypodontia and conical teeth - Onychodysplasia - Dry sparse hair	*WNT10A*	AD/AR	LoF	[78]
**Skeletal dysplasias:**						
Other skeletal dysplasias		- Osteopenia - Bilateral coxa valga deformity - Mild left radial and ulnar bowing - Broadening of metaphyses - Bilateral shortening of the great toes and thumbs	*WNT3A*	AR		[79]

Inh: Inheritance; Ref: References; AR: Autosomal recessive; AD: Autosomal dominant; DN: Dominant negative; LoF: Loss-of-function; GoF: Gain-of-function; ^1^ also known as: Osteosclerosis; Endosteal hyperostosis; Osteopetrosis type 1, van Buchem disease type 2. *: A 52kb deletion at 35 kb downstream of the *SOST* gene.

**Table 2 genes-13-00138-t002:** Mouse models of Wnt pathway co-receptor mutations.

Gene	Model	Tissue Expression	Phenotype			Comments	Refs.
			BM	BF	BR		
*Lrp5*	KO	Total KO	↓	↓	↓/= ^I^	^I^ ↓ in [80]; =in [81,82]	[80,81,82,83,84,85,86]
	cKO-Dermo1-Cre	EM	=	=	-		[86]
	cKO-Ocn-Cre	mature OB	↓	↓	=		[87]
	cKO-α1(I)-Col-Cre	OB	=	=	-		[82]
	cKO Rat 3.6Col1a1-Cre	OB	↓	-	-		[84]
	cKO-Dmp1-Cre	OCy	↓	-	-		[88]
	cKO-Vil1-Cre	ISC	↓/= ^II^	↓/= ^II^	-	^II^ ↓ in [82]; =in [88]	[82,88]
	cKO-Col21-Cre	CD and OB progenitors	↓	-	-		[89]
	KI-A214V or G171V	Total KI	↑	↑	=		[88]
	cKI-Dmp1-Cre A214V or G171V	OCy	↑	↑	=		[88]
	cKI-Prrx1-Cre A214V or G171V	MSC	↑	↑	=		[88]
	cKI-Vil1-Cre A214V	ISC	=	=	=		[88]
	cKI-Vil1-Cre G171V	ISC	↑/= ^III^	↑/= ^III^	=	^III^↑ in [82]; =in [88]	[82,88]
	cKI-1(I)-Col-Cre G171V	OB	=	=	=		[82]
	Tg Rat3.6Col1a1G171V	OB	↑				[90]
	cKI-Ctsk-Cre A214V or G171V	mature OC	↑	↑	↓ ^IV^	Offtargets in Ocy line ^IV^ Only in female mice	[91]
*Lrp6*	KO	Total KO	-	-	-	NV	[83,86,92]
	cKO-Dermo1-Cre	EM	=	=	=		[86]
	Ringelschwanz		↓	=	↑	p.R868W mutation	[93,94]
	cKO-Ocn-Cre	mature OB	↓	↓	=		[87]
	cKO-Col2a1-Cre	CD and OB progenitors	↓	-	-		[89]
*Lrp5* and *Lrp6*	KO-Lrp5 and Het KO-Lrp6	Total KO	↓	-	-	Severe limb abnormalities	[83]
	cKO-Dermo-Cre	EM	↓	↓	-	NV	[86]
	cKO-Ocn-Cre	mature OB	↓	↓	=	NV	[87]
	cKO-Col2a1-Cre	CD and OB progenitors	↓	-	-	NV	[89]
	cKO-Rank-Cre	OC precursors	↓	↓	↓		[95]
	cKO-Ctsk-Cre	mature OC	=	=	=		[95]
*Lrp4*	KO	Total KO				NV	[96,97,98]
	ECD		↓	↑	↑	Only ECD domain	[99]
	cKO-OCN-Cre	mature OB	↑	↑	↓		[96,100]
	cKO-Dermo1-Cre	OCy	↑	↑	↓		[100]
	cKO-LysM-Cre	Myeloid cells	=	=	=		[96]
	KI-Lrp4-R1170W	Total KI	↑	↑	=		[101]
	KI-Lrp4-R1170Q	Total KI	↑	↑	=		[102]
*Ror2*	KO	Total KO	-	-	-	NV	[8,103,104,105]
	Het KO	Het KO	↑	=	↓		[103]
	cKO-Rank-Cre	OC	↑	=	↓		[103]
	cKO-Ctsk-Cre	mature OC	↑	=	↓	Only in trabecular bone	[103,106,107]
*Krm1*	KO	Total KO	=	=	-		[108]
*Krm2*	KO	Total KO	=	=	-	In young mice	[108]
	KO	Total KO	↑	↑	-	In adult mice	[109]
	Tg Col1a1-Krm2	OB	↓	↓	↑		[109]
*Krm1* and *Krm2*	KO	Total KO	↑	↑	=		[108]
*Lgr4*	KO	Total KO	↓	↓/= ^V^	↑	^V^ ↓ [110,111]; = [112]	[110,111,112,113]
	cKO-LysM	Myeloid cells	↓	=	↑		[112]
*Lgr5*	KO	Total KO				NV	[114]
*Lgr6*	KO	Total KO	=	=	=		[115,116]

Ref: references; BM: bone mass; BF: bone formation BR: bone resorption; KO: knock-out; cKO: conditional KO; het KO: heterozygous KO; KI: knock-in; cKI: conditional KI; Tg: Transgenic; EM: embryonic mesenchyme; OB: osteoblast; OC: osteoclast; Ocy: osteocyte; CD: chondrocyte; ISC: intestinal stem cells; NV: not viable. Roman numerals: references to specific traits. ↑ increase; ↓ decrease; = not affected; - not available.

**Table 3 genes-13-00138-t003:** Mouse models of Wnt pathway ligand mutations.

Gene	Model	Tissue Expression	Phenotype			Comments	Refs.
			BM	BF	BR		
*Wnt1*	KO	Total KO	-	-	-	NV	[207,208]
	Het KO	Total Het KO	↓	=	=	Mild osteopenia	[207]
	Sway	Hypomorphic	↓	↓	=	G565del	[209,210]
	cKO-Prrx1-Cre	MSC	↓	↓	↑		[207]
	cKO-Runx2-Cre	OB	↓	-	-		[211]
	cKO-Dmp1-Cre	late OB and Ocy	↓	↓	-		[212]
	cKO-Lyz2-Cre	Monocytic lineage (OC included)	=	-	-		[211]
	Tg-Col1a1-tTA	OB, inducible	↑	↑	=		[211]
	Dmp1-Cre-R26 het Wnt1	Ov. in late OB and Ocy	↑	↑	=	HBM phenotype	[212]
*Wnt1* and *Lrp5*	Tg-Wnt1; Lrp5-KO	OB-targeted inducible Ov. of Wnt1 in LRP5-KO	↑	↑	=	Similar phenotype to LRP5+/+	[211]
*Wnt3a*	KO	Total KO	-	-	-	NV	[213,214]
*Wnt4*	KO	Total KO	-	-	-	NV	[215]
	cKO-Prrx1-Cre	MSC	↑ ^I^	-	-	^I^ femoral, in females	[216]
	Tg-Col2.3kb-Wnt4	OB	↑	↑	↓		[217]
*Wnt4* and *Wnt9a*	dKO	Total dKO	↓	-	-	Ectopic cartilage formation	[218]
*Wnt5a*	KO	Total KO	-	-	-	NV	[219,220]
	Het Wnt5a	Total Het KO	↓	↓	↓	Increased adipogenesis	[103,221]
	cKO-Osx-Cre	OB	↓	↓	↓		[103]
*Wnt5b*	KO	Total KO	-	-	-		[222]
*Wnt7a*	KO	Total KO	-	-	-	Abnormal limb development	[223]
*Wnt7b*	KO	Total KO	-	-	-	NV	[224]
	cKO-Dermo1-Cre	Skeletal progenitors	↓	-	-		[18]
	Tg-OSX-Cre;R26	Ov. in preOB	↑	↑	=	Little bone marrow space	[225]
	Tg-Col1-Cre;R26	Ov. in mature OB	↑	↑	=	Little bone marrow space	[225]
	Col2-Cre; R26	Ov. in OB and CD	↑	↑	-	Little bone marrow space	[210]
	Tg-Runx2-Wnt7b	Inducible total Ov.	↑	↑	=		[225]
	Osx-rtTA; tetO-Cre; R26	Inducible Ov. in OB	↑	↑	↑ ^II^	^II^ less than formation	[226]
*Wnt9a*	KO	Total KO	↓	-	-	NV, shortened long bones	[218]
*Wnt9a* and *Ror1*	Ror1hyp/hyp; Wnt9a-KO	Total Wnt9a KO, hypomorphic Ror1	↓	-	-	Deletion of Ror1 3rd exon, containing Ig-like domain	[166]
*Wnt9a* and *Ror2*	dKO	Total dKO	↓	-	-	Shortened long bones	[166,218]
*Wnt10a*	KO	Total KO	↓	↓	-	Impaired dental formation	[227,228,229]
*Wnt10b*	Wnt10b-KO	Total KO	↓	↓	-	Early age dependent	[230,231,232]
	Tg-FABP4-Wnt10b	Ov. in MSC	↑	-	-	Resistant to obesity and age/hormone- related bone loss	[232]
	Tg-Oc-Wnt10b	Ov. in OB	↑	↑	=		[231]
	Tg-Oc-Wnt10b	Ov. in mature OB	↑	↑	↑		[230,231]
*Wnt16*	KO	Total KO	↓	=/↓ ^III^	↑	in cortical bone ^II I^ ↓ [233]; ↑ = [234]	[233,234,235,236]
	cKO-Runx2-Cre	OB	↓	-	-	In cortical bone	[234]
	cKO-Dmp1-Cre	Ocy	=	-	-		[234]
	CAG-Cre-ER	Tamoxifen-inducible	↓	↓	↑	In cortical bone	[237]
	Tg-2.3Kb rat-Col1a1-Wnt16	OB	↑	=/↑ ^IV^	=/↓ ^V^	^IV^ = [238]; ↑ [239] ^V^ = [238]; ↓ [239]	[238]
	Tg-Dmp1-Wnt16	OCy	↑	↑	=		[240]

Ref: references; BM: bone mass; BF: bone formation; BR: bone resorption; KO: knock-out; cKO: conditional KO; het KO: heterozygous KO; Tg: Transgenic; Ov: overexpression; OB: osteoblast; OC: osteoclast; Ocy: osteocyte; CD: chondrocytes; NV: not viable. Roman numerals: references to specific traits.

**Table 4 genes-13-00138-t004:** Wnt pathway inhibitors in transgenic mouse models.

Gene	Model	Tissue Expression	Phenotype			Comments	Refs.
			BM	BF	BR		
*Dkk1*	KO	Total KO	-	-	-	NV	[306]
	Het KO	Het KO	↑	↑	=		[307,308]
	doubleridge	Hypomorphic	↑	-	-	Deletion of a regulatory region	[308,309]
	Rosa26-CreERT2	Tamoxifen-inducible at 7 weeks	↑	↑	↓ ^I^	^I^ Only the OC number	[310]
	cKO-OSX-Cre	Pre-OB	↑	↑	↓ ^I^	^I^ Only the OC number	[310]
	cKO-Dmp1-Cre	OCy	↑	↑	↓ ^I^	^I^ Only the OC number	[310,311]
	Tg-2.3 Kb rat-Col1a1	OB	↓	↓	=		[312,313]
*Wnt3* and *Dkk1*	dKO	Total Het KO	↑	↑	=	NV	[314,315,316]
*Dkk2*	KO	Total KO	↓	↓ ^II^	↑	Defects in mineralization; ^II^ Only BFR	[317]
*Sost*	KO	Total KO	↑↑	↑	=		[318,319]
	cKO-Prx2-Cre	Limb mesenchyme	↑↑	-	=		[320]
	cKO-2.3KbCol1a1-Cre	OB	↑	-	=		[320]
	Dmp1-Cre	OCy	↑	-	=		[320]
	Col10a1-Cre	CD	↑	-	=		[320]
	Tg-Dmp1-*SOST*	OCy	↓	↓	=		[321,322]
	Tg-Oc-*SOST*	OB	↓	↓	=		[323]
	Tg-*SOST*	Total Tg.	↓	↓	=	Insertion of ~158-kb from 3′-end of the *DUSP3* to *MEOX1*	[324]
*Sostdc 1*	KO	Total KO	↑	↑	↓	During the first 1.5 to 3 months of postnatal development	[325]
*Sostdc1* and *Lrp5*	dKO	Total KO	=	-	-		[325]
*Sfrp1*	KO	Total KO	↑	↑	=	Increased healing	[326,327,328]
	Tg-Sfrp1	Total Ov.	↓	-	-	Osteopenia	[329]
*Sfrp2*	KO	Total KO	=	=	=	Syndactyly	[326,330]
*Sfrp4*	KO	Total KO	↑^I II^/↓ ^IV^	↑ ^III^	↓ ^III^	^III^ in trabecular bone; ^IV^ in cortical bone	[61,106,331]
	Tg-Col1a1-Sfrp4	Ov. in OB	↓	↓	=		[332]
*Sfrp5*	KO	Total KO	-	-	-		[333]
*Sfrp4* and *Ror2*	Sfrp4-KO + Ror2-cKO	Total Sfrp4-KO; Ror2-cKO in OC	↑ ^V^	-	↓ ^V^	^V^ in trabecular bone	[106]
*Wif-1*	KO	Total KO	=	=	=		[334]
	Tg-2.3Col1a-Wif-1	Ov. in mature OB	=	=	=		[335]

Ref: references; BM: bone mass; BF: bone formation; BR: bone resorption; KO: knock-out; cKO: conditional KO; het KO: heterozygous KO; Tg: Transgenic; Ov: overexpression; OB: osteoblast; OC: osteoclast; Ocy: osteocyte; NV: not viable; BFR: bone formation rate. Roman numerals: references to specific traits.

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
