# Peer review of "Wnt Pathway Extracellular Components and Their Essential Roles in Bone Homeostasis"

_genes, 2022, doi:10.3390/genes13010138_

Round 1

Reviewer 1 Report

This review by Núria Martínez-Gil et al. provides an overview of the role of Wnt signaling in mammalian bone homeostasis by focusing on the co-receptors, ligands and inhibitors of the Wnt pathway. The authors did an excellent job overviewing the historical and recent findings on the studies from rodent models and human diseases associated with the genes covered by this manuscript. The figures and tables are well-designed. The review also provides a fair representation of the field's state with adequately cited references, which is timing and informative to the community. Specific comments were provided below on the areas that could be improved:

  1. The authors did an excellent job summarizing the results in the literature but rarely provided their own viewpoints on the results and the significance/limitation of the previous studies, especially on the contradictory results from different studies. e.g. line 200-204, global or conditional transgenic mice carrying human LRP5 HBM mutations show better bone quality, while cKI mice with the same mutations show a reduction in bone resorption; line 416-419, what is the potential in vivo function of LGR4 in BMSCs, in terms of osteogenic differentiation? Line 527-530, any comments on the different regulations of ALP by WNT3A? Authors’ viewpoints/comments/explanations are encouraged.
  2. Fig.1: legend of the extracellular domains (different shapes) of the receptors could be provided. Although it is provided in Fig.2, the shapes/colors differ between Fig.1 and Fig.2. e.g. low-density lipoprotein receptor domain class B domain of LRPs; LG-like domain/CRD domain of ROR2. What is the "POGR" label on the ER? Overall, Fig.1 is blurry.
  3. Information missing in the table and figure legends: Table.2: what dose “↓↓” indicate in line with ref [87]?  Figs.2-3: what is the difference between symbol “↓” and “⊥” in pointing out the location of mutations?  Fig.3: what do the “?” and the numbers mean?
  4. Please remove the color codes of the skeletal disease that are not shown in Figs.3 and 4.
  5. Some sentences are hard to understand. Please rephrase: line 96-97, what does “through PKC” mean? Line 410, what dose “administration” refer to? Line 615-617, what are the main points of the sentence “The Wnt5a-cKO….”   Please provide more details of previous studies covering the refs 133-143 (line 193-194).  The tense of the sentence in line 531-534 is not appropriate.
  6. Typos: line 82 “DLV”; line 117 “WIFI”; line 509 “the he”.

Author Response

This review by Núria Martínez-Gil et al. provides an overview of the role of Wnt signaling in mammalian bone homeostasis by focusing on the co-receptors, ligands and inhibitors of the Wnt pathway. The authors did an excellent job overviewing the historical and recent findings on the studies from rodent models and human diseases associated with the genes covered by this manuscript. The figures and tables are well-designed. The review also provides a fair representation of the field's state with adequately cited references, which is timing and informative to the community. Specific comments were provided below on the areas that could be improved:

The authors did an excellent job summarizing the results in the literature but rarely provided their own viewpoints on the results and the significance/limitation of the previous studies, especially on the contradictory results from different studies. e.g. line 200-204, global or conditional transgenic mice carrying human LRP5 HBM mutations show better bone quality, while cKI mice with the same mutations show a reduction in bone resorption; line 416-419, what is the potential in vivo function of LGR4 in BMSCs, in terms of osteogenic differentiation? Line 527-530, any comments on the different regulations of ALP by WNT3A? Authors’ viewpoints/comments/explanations are encouraged.

Thanks to the reviewer for encouraging us to provide our own opinion all of these points.

 Line 200-204: We have emphasized the relevance of the cKI in osteoclasts, and now the paragraph reads:

 To understand the effect of LRP5 on bone and the signaling pathways by which it acts, genetically modified mouse models have been generated (Table 2). Lrp5 total knock-out (Lrp5-KO) mouse models and conditional KOs (cKOs) in bone-related cells do not display bone alterations at birth but acquire a decreased BMD during postnatal development due to reduced bone formation [80,81,83–89] (Table 2). On the contrary, global or conditional LRP5 transgenic mice carrying the human LRP5 HBM p.Gly171Val or p.Ala214Val mutations reproduce the HBM phenotype with high rates of bone formation and better bone quality [88,90] (Table 2). Furthermore, conditional knock-in (cKI) female mice with the HBM mutations present only in osteoclasts, show a reduction in bone resorption, demonstrating that the in vivo effect of the HBM mutations is not only due to osteoblast stimulation but also to osteoclast inhibition [91] (Table 2).

 Line 416-419:

 In addition to the evidence from in vivo studies, many in vitro studies have been carried out demonstrating the positive effect of LGR4 on osteogenic differentiation [111,193,194], adipocyte and myocyte differentiation [111]. Although LGR4 is expressed within bone-marrow-derived mesenchymal stem cells (BMSCs) undergoing osteogenic differentiation, its expression is not correlated with any specific osteogenic marker but is maintained throughout the process, suggesting that LGR4 may be necessary to support osteogenic differentiation. of in vitro differentiation of osteoblasts of BMSCS [195].

 Line 527-530:

 In vitro, WNT3A plays an important role in the differentiation of MSCs into osteoblasts by the activation of the canonical pathway and the activation of PKC δ through the non-canonical pathway [18,250]. The expression of alkaline phosphatase (ALP), bone sialoprotein, osteocalcin and osterix is increased by WNT3A [18,248]. It is likely that in vivo, WNT3A concentration determines the fate of MSCs towards an adipogenic or osteogenic lineage through the activation of expression of ALP and other bone markers.

 Fig.1: legend of the extracellular domains (different shapes) of the receptors could be provided. Although it is provided in Fig.2, the shapes/colors differ between Fig.1 and Fig.2. e.g. low-density lipoprotein receptor domain class B domain of LRPs; LG-like domain/CRD domain of ROR2. What is the "POGR" label on the ER? Overall, Fig.1 is blurry.

 Thanks to the reviewer comment. We have homogenized the colors and shapes in Figures 1 and 2. We are referring to details of the different protein domains of Figure 1 to the legends of Figures 2-4, to avoid an extra-long legend in Fig.1 and redundant information in the legends. The label POGR is a typo and is PRCN: Porcupine. Thanks to the reviewer’s comment we have improved the quality of the image.

Information missing in the table and figure legends: Table.2: what dose “↓↓” indicate in line with ref [87]?  

 The cKO-Lrp5&Lrp6 mice from Table 2 displayed a highly decreased BM. However, we have changed the symbol to ↓ as it may lead to confusion.

Figs.2-3: what is the difference between symbol “↓” and “” in pointing out the location of mutations?  

 We have included the following explanation in figures 2-3: “↓ indicates a point mutation, encompasses more than one aminoacid position (a cluster of point mutations).”

Fig.3: what do the “?” and the numbers mean?

According to the HGVS (Human Genome Variant Server) the “?” indicates that the consequence, at the protein level, of a variant affecting the translation initiation codon (Met1) cannot be predicted. In the case of W9 of WNT10A, we have corrected the mistake and substituted the “?” for an “*”, indicative of a STOP-gain substitution.

Numbers below the structure show the aminoacid position in the peptide before postranslational modification.

Please remove the color codes of the skeletal disease that are not shown in Figs.3 and 4.

 All the skeletal diseases that do not appear in Figures 2, 3 and 4 were now deleted.

Some sentences are hard to understand. Please rephrase: line 96-97, what does “through PKC” mean? Line 410, what dose “administration” refer to? Line 615-617, what are the main points of the sentence “The Wnt5a-cKO….”   Please provide more details of previous studies covering the refs 133-143 (line 193-194).  The tense of the sentence in line 531-534 is not appropriate.

We have rephrased the following sentences:

 Line 96-97:

 “When the calcium concentration is increased, DAG activates PKC, which in turn can activate CDC42”

 Line 193-194:

 In addition, the LRP5 locus has been associated with bone mineral density (BMD) and risk of fracture in a plethora of genome wide association studies (GWAS) [133–143].

 Line 410:

 “Interestingly, the injection of a soluble form of the Lgr4 extracellular domain abrogated RANKL-induced bone loss in three mouse models of osteoporosis [112].”

 Line 531-534:

 “These results contradict those of Boland et al. ….”

 Line 615-617:

 “Wnt5a-cKO in osteoblasts also resulted in impaired bone formation, but also in impaired osteoclast formation and bone resorption as they failed to upregulate RANK expression in osteoclast precursors [103]”

Typos: line 82 “DLV”; line 117 “WIFI”; line 509 “the he”.

We have revised and corrected the typos in the manuscript.

Reviewer 2 Report

This is a well-written comprehensive review that covers mechanisms of Wnt signaling and how it regulates bone homeostasis.  The quality of manuscript is outstanding.  There are only minor suggestions.  One is to include the recent references regarding how Wnt1/Lrp5 regulate bone homeostasis:

https://doi.org/10.1002/jbmr.4303
https://doi.org/10.1038/s41413-021-00170-0
https://doi.org/10.1038/s41413-021-00152-2
https://doi.org/10.1177%2F00220345211012386

Another suggestion is to provide a concluding chapter that discusses what the future holds for studying Wnt signaling in bone.  The current review abruptly ends with WIF-1.  After reading such a comprehensive review, a reader is likely to yearn for some summary and what the next frontier is, whether further dissecting how disrupted Wnt signaling affects bone, which in return indirectly leads to endocrine abnormality, or the effect of aging.  Such a concluding chapter will further increase the impact of this manuscript. 

Author Response

This is a well-written comprehensive review that covers mechanisms of Wnt signaling and how it regulates bone homeostasis.  The quality of manuscript is outstanding.  There are only minor suggestions.  One is to include the recent references regarding how Wnt1/Lrp5 regulate bone homeostasis:

Leanza2021 - https://doi.org/10.1002/jbmr.4303
Vollersen2021 - https://doi.org/10.1038/s41413-021-00170-0
Ko2021 - https://doi.org/10.1002/jor.25217
Liu2021 - https://doi.org/10.1038/s41413-021-00152-2
Nottmeier2021 - https://doi.org/10.1177%2F00220345211012386

Many thanks to the reviewer for these 5 references that show such interesting results. Our review includes references until beginning of 2021, which is during the time that it has been written, that is why these works had not been included. Despite this, we have included 2 of the references suggested here in the final paragraph to illustrate the new recent advances in the field of Wnt's pathway and bone homeostasis. These three are the works of Leanza et al. (now Ref #465), and Liu et al. (now Ref #466).

Another suggestion is to provide a concluding chapter that discusses what the future holds for studying Wnt signaling in bone.  The current review abruptly ends with WIF-1.  After reading such a comprehensive review, a reader is likely to yearn for some summary and what the next frontier is, whether further dissecting how disrupted Wnt signaling affects bone, which in return indirectly leads to endocrine abnormality, or the effect of aging.  Such a concluding chapter will further increase the impact of this manuscript. 

Thanks to the reviewer for encouraging us to include a final paragraph, as follows:

Future and conclusions

The Wnt pathway is one of the most studied signaling pathways within the scientific community due to its involvement in different mechanisms ranging from development and cancer to bone homeostasis. During all this time, numerous advances have been made in the knowledge of this pathway, but due to its extreme complexity there are still aspects to be identified, such as the differences in the performance in space (specific tissue) and time (from development to ageing). Specifically, for bone health, understanding these mechanisms is of great interest for the identification of therapeutic targets that allow treating very common bone diseases such as osteoporosis or very severe conditions such as osteogenesis imperfecta. This is represented by the recent anti-sclerostin antibody treatment that is providing very promising results in clinical management [422-427]. In the last year, other new advances in this topic have been published including the positive effects of LRP5 (HBM mutation or overexpression) in conditions such as hyperglycemia and breast cancer, respectively [465, 466] illustrating the relevance of cross-talk between bone and other tissues and the importance of studying the Wnt pathway at physiological level beyond bone. Other areas of future interest should be the ageing bone, osteoarthritis, and the relationship with vascular calcification. For this reason, a comprehensive study of the classic elements of the Wnt pathway (e.g. LRP5, DKK1, WNT1) as well as other less studied extracellular elements (e.g. KREMEN, SFRP4), which may be analyzed through the development and characterization of appropriate animal models, can provide us with new useful therapeutic targets.